# Sensory nerve niche regulates mesenchymal stem cell homeostasis via FGF/mTOR/autophagy axis

Fei Pei [1,2], Li Ma[1], Junjun Jing[1], Jifan Feng[1], Yuan Yuan[1], Tingwei Guo[1], Xia Han[1], Thach-Vu Ho[1], Jie Lei[1], Jinzhi He[1], Mingyi Zhang[1], Jian-Fu Chen [1] & Yang Chai [1] ✉

Mesenchymal stem cells (MSCs) reside in microenvironments, referred to as niches, which provide structural support and molecular signals. Sensory nerves are niche components in the homeostasis of tissues such as skin, bone marrow and hematopoietic system. However, how the sensory nerve affects the behavior of MSCs remains largely unknown. Here we show that the sensory nerve is vital for mesenchymal tissue homeostasis and maintenance of MSCs in the continuously growing adult mouse incisor. Loss of sensory innervation leads to mesenchymal disorder and a decrease in MSCs. Mechanistically, FGF1 from the sensory nerve directly acts on MSCs by binding to FGFR1 and activates the mTOR/autophagy axis to sustain MSCs. Modulation of mTOR/autophagy restores the MSCs and rescues the mesenchymal tissue disorder of *Fgfr1* mutant mice. Collectively, our study provides insights into the role of sensory nerves in the regulation of MSC homeostasis and the mechanism governing it.

Stem cells give rise to different cell lineages and participate in tissue homeostasis during embryogenesis, postnatal development, injury repair, and throughout the entire life. Stem cells are in part characterized by their self-renewal in a complex multidimensional environment, known as the stem cell niche[1]. The niche environment provides stem cells with anatomical and functional support. While we have accumulated substantial knowledge of the signals regulating stem cell quiescence and activation, the key components of the stem cell niche remain to be fully elucidated, and the study of the niche remains difficult because of its complexity and dynamic change.

The mouse incisor is an organ that undergoes continuous self-renewal, similar to ectodermal tissues, such as hair follicles, intestinal epithelium, skin, and nails. Significantly, the mouse incisor serves as an ideal model to investigate the molecular regulation of mesenchymal stem cells (MSCs) as they continuously fuel this self-renewal to maintain tissue homeostasis and participate in injury repair[2]. Taking advantage of this model, we have investigated the regulatory

mechanisms of stem cell maintenance, tissue homeostasis, regeneration, and niche components. Recent studies have explored certain components of the niche environment, including stem cells and daughter cells, stromal cells, and external cues like blood vessels, nerves, and immune cells[3,4]. For example, in intestinal crypts, Paneth cells regulate intestinal stem cell function by secretion of WNT and EGF[5,6]. Stromal cells secrete factors such as WNT, Notch, and BMP to maintain epithelial stem cells[3]. Runx2+ cells are niche cells involved in mesenchymal homeostasis via IGF signaling, and IGF-WNT signaling participates in MSC–transit amplifying cell (TAC) interaction in the incisor[7,8]. A comprehensive understanding of the role of niche components is important for our investigation of the regulatory mechanism of tissue homeostasis.

Adult organs are typically highly innervated, and peripheral nerves are an important component of the tissue environment. Nerves are known to be involved in the stem cell microenvironment. For example, sympathetic nerves in the bone marrow regulate bone

[1]Center for Craniofacial Molecular Biology, University of Southern California, 2250 Alcazar Street, CSA 103, Los Angeles, CA 90033, USA. [2]The State Key Laboratory Breeding Base of Basic Science of Stomatology (Hubei-MOST) & Key Laboratory of Oral Biomedicine Ministry of Education, School and Hospital of Stomatology, Wuhan University, 430079 Wuhan, China. ✉e-mail: ychai@usc.edu

marrow stem cell (BMSC) maintenance, proliferation, and differentiation. Depletion of sympathetic nerves in bone marrow impairs the differentiation capacity of BMSCs and compromises the hematopoietic stem cells (HSCs)[9–11]. In the hair follicle, the hyperactivation of sympathetic nerves leads to the hyperproliferation of melanocyte stem cells, which causes the eventual elimination of these stem cells from the niche and results in hair discoloration[12]. Nerves have also been shown to be indispensable for tissue repair and regeneration, such as digit regeneration[13,14] and skull and mandibular repair[15,16]. Recently, sensory nerves have been found to exist in various stem cells microenvironments, such as bone[17], skin[18], lung[19], and cancer[20]. Deletion of sensory nerves decreases bone mass[21], and sensory nerves control bone tissue homeostasis in coordination with sympathetic nerves[22]. Sensory nerves sense signals from osteoblasts through the PGE2/EP4 axis and tune down sympathetic nerves to regulate bone formation[17]. However, sensory nerves not only sense stimuli and send information to the central nervous system, but also secrete neuropeptides and other factors, such as calcitonin gene-related peptide (CGRP), substance P, and Sonic hedgehog (Shh)[23]. Study has shown that neuron-derived semaphoring 3A (Sema3A), as a diffusible axonal chemorepellent, indirectly modulate bone remodeling[21]. In hair follicles[24] and adult mouse incisor[2], Shh secreted from sensory nerves is required to support stem cell maintenance. In calvarial bone, sensory nerves secrete FSTL1 to modulate mesenchymal progenitor cells during development[25], and sensory nerves are crucial for calvarial and mandibular bone repair[15,16]. A recent study has shown that sympathetic nerves indirectly regulate HSCs through their niche, whereas nociceptor neurons secrete calcitonin gene-related peptide (CGRP) to act directly with the receptor on HSCs to promote HSC mobilization[26]. Studies on how factors derived from sensory nerves target stem cells are limited, which severely limits our understanding of the mechanism of the regulatory role of sensory nerves in stem cell niche and tissue homeostasis.

The stem cell properties of quiescence, self-renewal, multipotency and differentiation in adult tissues suggest that stem cells may regulate the turnover of proteins and organelles. Autophagy is essential to stem cell maintenance with rigorous modulation of stem cell properties, as recent studies have shown[27]; it plays pivotal roles in embryonic and adult stem cells, and deficiency of autophagy can cause stem cell exhaustion, cell death, or senescence[28]. Single-cell RNA sequencing has revealed that autophagy-related genes are highly expressed during fetal HSC formation[29]. HSCs have high levels of basal and induced autophagy, which suggests that autophagy is important for HSCs to function normally. Deletion of Atg7 in the hematopoietic system in mice results in an increase in reactive oxygen species, DNA damage, and proliferation, ultimately causing blood malignancies[30]. Stem cell disorders lead to numerous conditions, such as defective organ development, aging, cancers, and impaired injury repair. Therapeutic approaches targeting autophagy have been implicated in preserving stem cell function[28,31,32].

In this study, we discovered that the sensory nerve is an essential stem cell niche component in the adult mouse incisor. Sensory denervation leads to MSC loss and mesenchymal tissue disorder. The bridge connecting the sensory nerve to MSCs is FGF signaling. We found that the FGF1 ligand is secreted from sensory neurons and acts directly on the specific receptor FGFR1 on MSCs to regulate the retention, survival, and differentiation of MSCs in the adult mouse incisor. Deletion of FGF signaling leads to a decrease in the number of MSCs and TACs and disrupts MSCs and tissue homeostasis. Furthermore, once MSCs receive the FGF1 signal from sensory nerves, they activate p-JNK signaling, which regulates mTOR-dependent autophagy to modulate MSC maintenance. Moreover, the re-establishment of autophagy function can restore MSCs and disordered mesenchymal tissue in *Fgfr1* mutant mice. This discovery illustrates the direct regulation of MSCs by sensory nerves through interaction between non-

neuropeptides, specifically FGF1 and receptor FGFR1 on the MSCs, providing new insight into how sensory nerves regulate tissue homeostasis. Our findings establish that sensory nerves regulate MSCs through an FGF/mTOR/autophagy axis, which suggests a possible future approach for tissue regeneration.

## Results

### The sensory nerve niche supports mesenchyme tissue homeostasis

The incisor is a highly innervated organ, in which nerves accompany arteries and veins to form the neurovascular bundle[2]. We detected the spatial distribution of nerves in the incisor with wholemount neurofilament staining, through which we produced a 3D image of nerves in the incisor (Supplementary Movie 1). Nerve fibers were enriched in the proximal end of the incisor, in which nerve terminals could be detected (Supplementary Movie 1). To investigate the nerve distribution in the adult mouse incisor, we used neurofilament staining to detect total nerve fibers and different nerve markers to verify the composition of the nerves in the adult mouse incisor. Neurofilament (NF) was highly expressed in the incisor (Fig. 1a). Sensory nerves, identified by calcitonin gene-related peptide (CGRP) (Fig. 1b), were co-stained with neurofilament (Fig. 1d). Very sparse sympathetic nerves, positive for tyrosine hydroxylase (TH), were identified in the proximal follicle and pulp of the incisor (Fig. 1c). Parasympathetic nerves, stained with choline acetyltransferase (ChAT), could not be detected in the incisor. These findings indicate that the incisor was predominantly innervated by sensory nerves.

To study the role of sensory nerves in regulating mouse incisor tissue homeostasis, we used *Advillin^CreER* mouse model to genetically target sensory nerves. Specifically, *Advillin^+* cells were colocalized with pan-neuronal marker TUJ1 in the trigeminal ganglion of *Advillin^CreER;tdTomato* mice (Supplementary Fig. 1a). We also detected the colocalization of tdTomato and S100 in the inferior alveolar nerve (IAN) and nerve fibers in the incisor (Supplementary Fig. 1b). This confirmed that *Advillin^CreER* efficiently targets sensory nerves in the incisor. To test whether the sensory nerve is indispensable for incisor tissue homeostasis, we generated *Advillin^CreER;RosaDTA* mice. The neurons positive for P92 significantly decreased in the trigeminal ganglion of *Advillin^CreER;Rosa-DTA* mice in comparison to those of controls (Supplementary Fig. 1c, e), and nerve fibers in the incisor could not be detected one month after TMX induction in *Advillin^CreER;Rosa-DTA* mice (Fig. 1l, n). This confirmed the efficient deletion of sensory nerves in *Advillin^CreER;Rosa-DTA* mice. One month after injection, the dental pulp had significantly narrowed in the incisors of *Advillin^CreER;Rosa-DTA* mice (Supplementary Fig. 1g, j). The histological analysis showed flat and unpolarized pre-odontoblasts in the proximal end of the incisor in the control group, whereas polarized pre-odontoblasts could be seen in the proximal end of the incisor along with pre-dentin formation in *Advillin^CreER;Rosa-DTA* mice (Supplementary Fig. 1h). The expression of *Dspp*, an odontoblast differentiation marker, was present closer to the proximal end of the incisor (the incisor epithelial cervical loop) in *Advillin^CreER;Rosa-DTA* mice (Supplementary Fig. 1i, k). Three months after the TMX injection, abnormal dentin formation led to a narrowed pulp cavity (Fig. 1e, h). The dentin deposited at the proximal end of the incisor was increased in *Advillin^CreER;Rosa-DTA* mice when compared to control samples (Fig. 1f). The expression of *Dspp* was even closer to the proximal end of the incisor (Fig. 1g, i). These findings suggest that mesenchymal homeostasis was disrupted following the loss of sensory nerve. Since the homeostasis of mesenchymal tissue in the incisor is closely related to the growth rate of the tooth, we decided to investigate the incisor growth rate. Specifically, we notched the enamel one month after tamoxifen induction and analyzed the notch movement (Fig. 1j). There was no obvious change between control and mutant mice on day 3 and day 6. But by day 14, the movement of the notch was significantly slower in

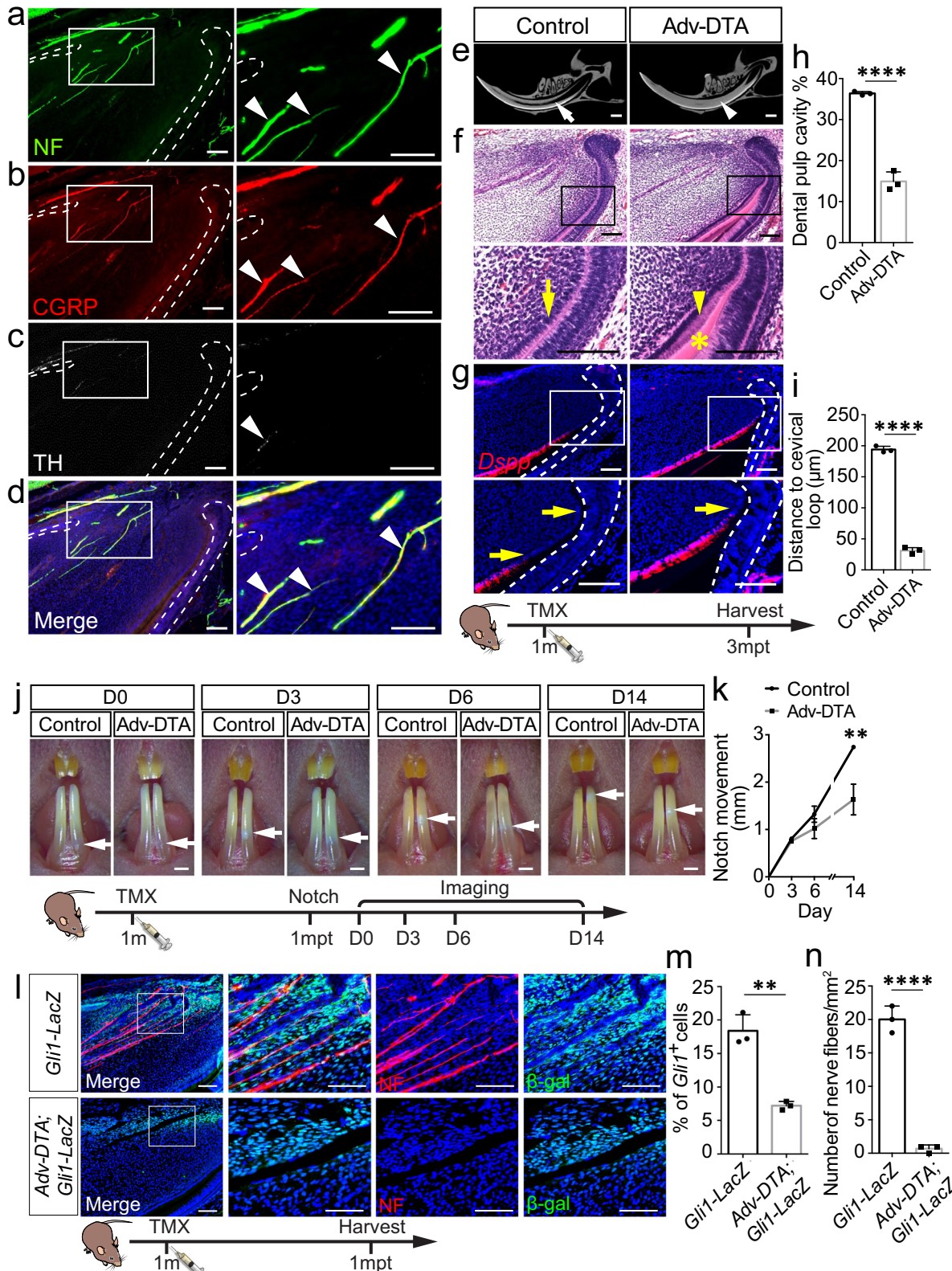

*Advillin^{CreER};Rosa-DTA* mice (Fig. 1j, k). These findings suggest that the sensory nerve is indispensable for incisor growth and tissue homeostasis, deletion of which could lead to a reduced growth rate of the incisor and abnormal dentin formation.

Since MSCs play a crucial role in mesenchymal tissue homeostasis, we further explored the relationship between the sensory nerve and stem cells in *Gli1-LacZ* mice. We found that *Gli1+* cells were located close to nerves in the proximal mesenchyme of the incisor (Fig. 1l), indicating a close relationship and potential interaction between sensory nerves and MSCs. To elucidate whether the sensory nerve affects MSCs' fate, we depleted sensory nerves by TMX injection of *Advillin^{CreER};Rosa-DTA;Gli1-LacZ* mice and found that the number of *Gli1+* cells had significantly decreased at one month post-TMX induction after sensory nerves were compromised efficiently (Fig. 1l, m). All

**Fig. 1 | Sensory nerve is essential for mesenchyme tissue homeostasis in adult mouse incisors. a–d** Distribution of nerves in the adult mouse incisor. **a** Neurofilament (NF) is highly expressed in the nerve fibers in the proximal end of the incisor. **b** Sensory nerves in the incisor were identified by calcitonin gene-related peptide (CGRP). **c** Sympathetic nerves were stained with tyrosine hydroxylase (TH). **d** CGRP⁺ sensory nerve co-stained with Neurofilament. White arrowhead points to nerves. **e–i** Depletion of sensory nerves leads to abnormal dentin formation. **e** CT scanning of control and *Advillin^CreER;Rosa-DTA* mice 3 months after tamoxifen induction. White arrow points to dental pulp cavity; white arrowhead points to narrowed pulp cavity. **f** Histological analysis of control and *Advillin^CreER;Rosa-DTA* mice. Yellow arrow points to pre-odontoblast; yellow arrowhead points to abnormal pre-odontoblast; asterisk points to abnormal dentin formation. **g** *Dspp* expression appeared closer to the proximal end of the incisor in *Advillin^CreER;Rosa-DTA* than in controls. Yellow arrow points to the distance between the bending point of the cervical loop and the initiation of odontoblast differentiation. Schematic at the bottom indicates the induction protocol. mpt, month post-tamoxifen injection. **h** Quantification of dental pulp

cavity percentage in control and mutant mice. *P* < 0.0001. **i** Quantification of distance of *Dspp*⁺ cells to cervical loop in control and mutant mice. *P* < 0.0001. **j** Deletion of sensory nerve resulted in slower growth of the incisor. Notch movement was observed at day(D) 3, D6 and D14 in control and *Advillin^CreER;Rosa-DTA* mice. White arrow points to the notch location. Schematic at the bottom indicates induction protocol. mpt, month post-tamoxifen injection. **k** Quantification of notch movement in control and mutant mice. *P* = 0.0042. **l** Nerves labeled with neurofilament and *Gli1*⁺ cells stained with β-gal in *Gli1-LacZ* mice and *Advillin^CreER;Rosa-DTA;Gli1-LacZ* mice one month after tamoxifen induction. **m** Quantification of *Gli1*⁺ cell in control and mutant mice. *P* = 0.0015. **n** Quantification of nerve fibers in control and mutant mice, *P* < 0.0001. For **h**, **l**, **k**, **m**, and **n**, *n* = 3 and each data point represent one animal, with unpaired Student's *t*-test performed. All data are expressed as the mean ± SD. Source data are provided as a Source Data file. **P* < 0.01, ***P* < 0.0001. Each experiment was repeated independently three times. White dotted line outlines the cervical loop. Adv-DTA: *Advillin^CreER;Rosa-DTA*. Scale bars, **j**, 2 mm; others, 100 μm.

these findings demonstrate that the sensory nerve is a crucial stem cell niche that modulates stem cell maintenance, mesenchymal tissue homeostasis, and incisor growth. Depletion of the sensory nerve led to the loss of MSCs and disrupted mesenchymal tissue homeostasis.

## Sensory nerve regulates mesenchymal cells in the incisor through FGF signaling

To investigate how the sensory nerve regulates mesenchymal tissue homeostasis, we performed single-nucleus (sn) RNAseq of the trigeminal ganglion in adult mice (Fig. 2a). Twelve clusters were identified in the trigeminal ganglion including sensory neurons, satellite glial cells, Schwann cell progenitors, and myelinating/non-myelinating Schwann cells. Sensory neurons constituted the majority of the trigeminal ganglion (Fig. 2b, c). To explore the interaction between the sensory nerve and cells in the incisor, we integrated the snRNAseq data from the trigeminal ganglion with scRNAseq data collected for our previous study of the adult incisor[7]. The Seurat object was imported into CellChat to analyze the significant signaling pathways involved in the interaction between cells in the trigeminal ganglion and incisor. We analyzed the most prominent signaling from the trigeminal ganglion. MPZ, CADM, CNTN, FGF, L1CAM, and THY1 were outgoing signals from trigeminal ganglion; of these, MPZ, CADM, and CNTN signalings targeted the sensory neurons and glial cells (Supplementary Fig. 2a). This means they have an autocrine role in regulating nerves. FGF signaling from the trigeminal ganglion was received mainly by cell clusters in the incisor, such as proximal mesenchymal cells, dental follicles, TACs, epithelial cells, and odontoblasts (Supplementary Fig. 2a). This suggested that FGF signaling was the most significant signaling involved in the interaction between the sensory nerve and the incisor. Network analysis of FGF signaling further showed that sensory neurons in the trigeminal ganglion were the most prominent source of FGF ligands acting on proximal mesenchymal cells (MSC region) in the adult mouse incisor (Fig. 2d). FGF signaling, therefore, appeared to be an important bridge that connects sensory nerves to proximal mesenchymal cells.

There are six subfamilies of fibroblast growth factors (FGFs) including FGF1-FGF10 and FGF16-FGF23[33]. To investigate which FGF ligand is involved in the interaction, we detected the expression of canonical FGFs in the trigeminal ganglion. We found that *Fgf1* was highly expressed in sensory neurons, while other *Fgfs* were expressed at a low level (Fig. 2e). Apart from the main source of FGF signaling from sensory nerves, some FGF signaling from TACs and odontoblasts in the incisor had an auxiliary effect on mesenchymal cells in the incisor (Fig. 2d). We also detected FGF ligands in the adult mouse incisor to check the local FGF signaling (Supplementary Fig. 3a, b). Sparse *Fgf1* was detected in the epithelium. *Fgf3*, *Fgf8*, and *Fgf10* were expressed in dental mesenchymal cells (Supplementary Fig. 3b–f, i, j).

Expression of *Fgf9* was also found in the epithelium (Supplementary Fig. 3b, g, h). These findings are consistent with previous studies, which showed that FGF signaling (FGF3 and FGF10) in the mesenchyme is crucial for the dental epithelium[34,35]. To further explore the main source of FGF signaling from the sensory nerve, we confirmed that *Fgf1* transcripts were confined to the neuronal cell bodies within the trigeminal ganglion (Fig. 2f, g) but not in the incisor mesenchyme (Fig. 2h, i). Significantly, we detected the presence of FGF1 protein in both neuronal cell bodies in the trigeminal ganglion and the proximal mesenchymal region of the incisor (Fig. 2j–m). These findings suggest that neurons in the trigeminal ganglion may secrete FGF1 to act upon the incisor mesenchyme.

To further test that trigeminal sensory neuron innervating the incisor mesenchyme secrete FGF1, we performed retrograde tracing with CTB-488 (Supplementary Fig. 2b). It showed that the neurons innervating the incisor mesenchyme originated exclusively from the V3 branch of the trigeminal nerve (Supplementary Fig. 2c, d). Then we detected *Fgf1* expression in V3 and found that the CTB-488⁺ neurons were all colocalized with *Fgf1* (Supplementary Fig. 2f, j, h, l). Since sensory neurons in the trigeminal ganglion are heterogeneous, based on our and other studies[36], we explored which type of sensory neuron was responsible for FGF1 secretion. We found that most CTB-488⁺ neurons releasing FGF1 also expressed *Gfra2* (Supplementary Fig. 2e–l), which suggested that mechanoreceptor (*Gfra2*⁺ neurons) predominantly control the release of FGF1 in the adult mouse incisor. These studies clearly demonstrated that the sensory neurons are the source of FGF1 in the mouse incisor. Taken together, these findings illustrate that FGF signaling is important for the interaction between sensory nerves and the proximal mesenchyme of the incisor, and FGF1 secreted from sensory neurons may regulate mesenchymal tissue homeostasis.

## Nerve-derived FGF1 is crucial for MSC maintenance

Since *Advillin^CreER;Rosa-DTA* mice target all the sensory nerves, we performed denervation of the inferior alveolar nerve, V3 of the trigeminal ganglion responsible for incisor innervation, to specifically detect the role of incisor innervation in MSC maintenance. One month after denervation, the nerve fibers decreased obviously and the number of MSCs was reduced (Fig. 3a, k, l). To further confirm that FGF1 from the sensory nerve is essential for MSC maintenance, we cultured incisor explants with IgG or neutralizing FGF1 antibody-loaded beads (Fig. 3b). The *Gli1*⁺ cells were maintained with the IAN surrounding the proximal end of the incisor for 3 days of culture with IgG loaded beads. However, the number of *Gli1*⁺ cells decreased when we added the neutralizing FGF1 antibody-loaded beads on the other side of the IAN (Fig. 3c, m). This suggested that sensory nerve-derived FGF1 may sustain MSCs.

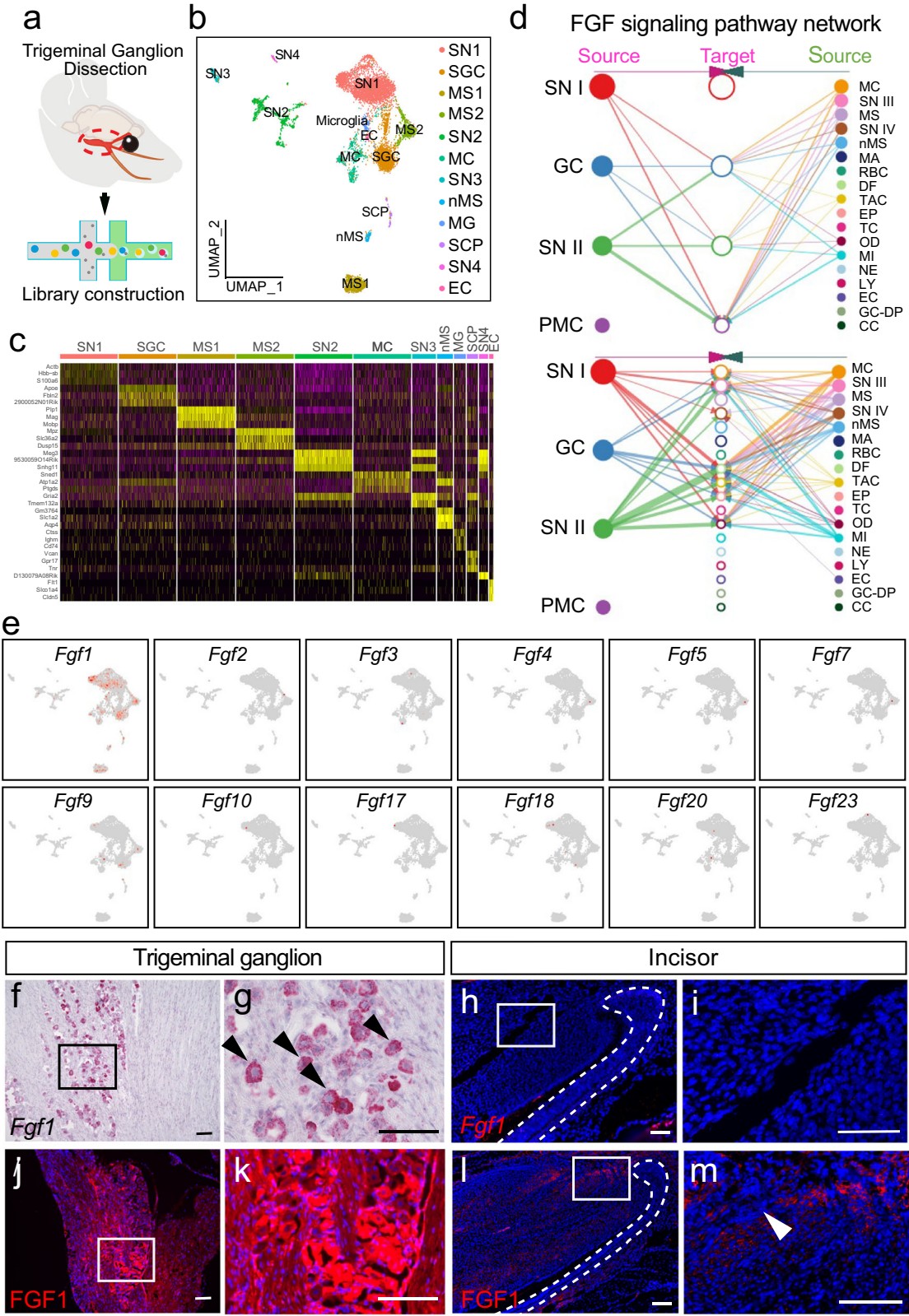

To verify this in vivo, we performed *Fgf1* shRNA lentiviral injection to V3 of the trigeminal ganglion to decrease *Fgf1* expression in sensory neurons innervating the incisor with *Gli1-LacZ* mice (Fig. 3d). *Fgf1* expression decreased in V3 of the trigeminal ganglion, while V1 and V2 showed no significant change (Fig. 3e, f, n), which confirmed the efficient and specific reduction of *Fgf1* in V3 of the trigeminal ganglion. One month after *Fgf1* shRNA lentiviral injection, we found

that the incisor dentin pulp cavity was narrowed (Fig. 3g, o), with dentin deposition at the proximal end of the incisor (Fig. 3h), which was similar to the phenotype seen in *Advillin^{CreER};Rosa-DTA* mice (Supplementary Fig. 1h). The expression of *Dspp* was closer to the proximal end of the incisor (Fig. 3i, p). The number of MSCs decreased in *Fgf1* shRNA lentivirus-treated mice (Fig. 3j, q). These findings demonstrated that FGF1 derived from the sensory nerve is

**Fig. 2 | Sensory neurons secrete FGF1 to regulate mesenchymal cells in the incisor. a** The schematic of the experimental procedure for snRNA-Seq. **b** 12 clusters across 8679 cells from all cell types in the trigeminal ganglion on a UMAP visualization. SN1–4, sensory neuron type 1–4; SGC, satellite glial cell; MS I-II, myelinating Schwann cell type I-II; nMS non-myelinating Schwann cell; SCP Schwann cell progenitor; MC meningeal cell, EC endothelial cell. **c** Expression of marker genes of each cell subtype in the mouse trigeminal ganglion. **d** Hierarchical plot shows the inferred intercellular communication network for FGF signaling. SNI-IV sensory neuron type I–IV, GC glia cell, PMC proximal mesenchymal cell, MC meningeal cell, MS myelinating Schwann cell, nMS non-myelinating Schwann cell, MA macrophage, RBC red blood cell, DF dental follicle cell, TAC TA cell TC T cell,

OD odontoblast, MI microglia, NE neutrophil, LY lymphocyte, EC endothelial cell, GC-DP glia cell in dental pulp, CC cycling cell. **e** Expression levels of canonical FGFs for 12 cell types in the trigeminal ganglion. *Fgf1* is highly expressed in sensory neurons, while other FGFs show little expression. **f** and **g** The expression of *Fgf1* in the trigeminal ganglion. **h** and **i** *Fgf1* is undetectable in the mesenchyme of the incisor. **j** and **k** Protein level of FGF1 in the trigeminal ganglion. **l** and **m** Protein level of FGF1 in the incisor. Black arrowheads point to neuron bodies secreting *Fgf1*; white arrowhead points to FGF1 expression in the incisor. White dotted line outlines the cervical loop. Each experiment was repeated independently three times. Scale bars, 100 μm.

---

crucial for the maintenance of MSCs and mesenchymal tissue homeostasis.

## FGF1 secreted from sensory nerves directly acts on MSCs via FGFR1

After determining that FGF1 secreted from sensory neurons targets the proximal mesenchyme of the incisor and sustains MSCs, we sought to determine which mesenchymal cell population FGF1 targets. We found that FGF1 was expressed in the proximal end of the incisor and surrounding *Gli1*+ cells as shown in *Gli1-LacZ* mice (Fig. 4a). To verify the role of FGF1 in regulating MSCs, we used *Gli1-LacZ* mice to perform incisor explant culture (Supplementary Fig. 4a). In incisor explants, *Gli1*+ cells could be detected in the proximal region but decreased in number after culturing for 3 days (Supplementary Fig. 4b, c). In explants with the IAN surrounding the proximal end of the incisor, the *Gli1*+ cells were partially maintained. Similar result was also observed in explants treated with recombinant FGF1 (Supplementary Fig. 4c). These experiments established an interesting parallel to our finding of decreased MSCs following the loss of sensory nerve in vivo. Collectively, our study suggests that FGF1 may act on MSCs in the incisor.

FGF ligands exert their different functions by binding to and activating FGFR family members[33]. To test our hypothesis, we used CellChat to analyze the contributions of ligand-receptor pairs, which revealed that FGF signaling is dominated by FGF1 ligand and FGFR1 receptor (Fig. 4b). We next detected the in vivo expression of different FGFRs in the incisor. Consistent with CellChat analysis, FGFR1 was the dominant receptor in the MSC region. FGFR1 was expressed in some *Gli1*+ cells in the proximal mesenchyme where FGF1 was enriched, as well as in the epithelium (Fig. 4c). However, *Fgfr2*, *Fgfr3*, and *Fgfr4* were undetectable in *Gli1*+ cells (Supplementary Fig. 4d–f). *Fgfr2* was detected in the follicle and dental epithelium (Supplementary Fig. 4d). *Fgfr3* was detected in the sub-odontoblast layer and partial pulp cells (Supplementary Fig. 4e). *Fgfr4* was not expressed in the mesenchyme or the epithelium of the incisor (Supplementary Fig. 4f). These findings suggested that FGF1 dominantly targets MSCs in the incisor. To further test the interaction between FGF1 and FGFR1, we collected proximal incisor mesenchyme and confirmed the binding between FGF1 and FGFR1 with co-immunoprecipitation (Fig. 4d). We also cultured MSCs isolated from *Gli1CreER;tdTomato* mice and detected the expression of *Fgfr1* in all tdTomato+ cells (Fig. 4e). These results indicated that FGF1 binds with FGFR1 on MSCs to potentially regulate MSC fate through FGF signaling.

To test this hypothesis, we generated *Gli1CreER;Fgfr1fl/fl* mice to delete FGFR1 in *Gli1*+ cells. We validated that FGFR1 was efficiently deleted by tamoxifen induction in *Gli1CreER;Fgfr1fl/fl* mice (Supplementary Fig. 4g). Overall, the phenotypes of which were similar to those seen in *AdvillinCreER;RosaDTA* mice. Three months after induction, we found abnormal dentin deposition and narrowed dental pulp in the *Fgfr1* mutant mice (Fig. 4f, i). Similarly, odontoblasts were arranged in a disorderly fashion with abnormal dentin accumulation in the proximal region of *Gli1CreER;Fgfr1fl/fl* incisors (Fig. 4g). *Dspp* expression was found closer to the proximal end of the incisor in the *Fgfr1* mutant mice (Fig. 4h, j). The growth rates of *Gli1CreER;Fgfr1fl/fl* incisors were decreased

in comparison to the controls through notch movement analysis (Fig. 4k, l). To further test the incisor tissue repair rate, we also injured incisors by clipping them in control and *Fgfr1* mutant mice, and the results showed decreased incisor repair capacity in *Gli1CreER;Fgfr1fl/fl* mice (Supplementary Fig. 4h, i). These findings suggest FGF signaling in MSCs regulates adult mouse incisor growth, homeostasis, and injury repair.

Since *Fgfr1* is also expressed in the dental epithelium, we generated *K14rtTA;tetOCre;Fgfr1fl/fl* mice to test whether loss of *Fgfr1* in the epithelium would also lead to mesenchymal defects. There was no obvious change in dentin formation between control and *K14rtTA;tetO-Cre;Fgfr1fl/fl* mice based on microCT analysis (Supplementary Fig. 5a, b, q). The odontoblast morphology and differentiation showed no obvious change in histological analysis or *Dspp* expression (Supplementary Fig. 5c–h, r). These findings demonstrated that FGFR1 in the mesenchyme, rather than the epithelium, is responsible for the maintenance of tissue homeostasis in the adult mouse incisor. Since we found expression of *Fgfr2* in the follicle of the incisor, we also generated *Gli1CreER;Fgfr2fl/fl* mice to assess the potential contribution of FGFR2 in regulating incisor tissue homeostasis. Based on our analysis, there was no obvious change in dentin formation between control and *Gli1CreER;Fgfr2fl/fl* mice one month after TMX induction (Supplementary Fig. 5i, j, s). The odontoblast arrangement and differentiation were also unaffected (Supplementary Fig. 5k–p, t). This suggests that FGFR2 is not the binding receptor for FGF1 in modulating mesenchymal tissue homeostasis in adult mice. Rather, FGF1 secreted from the sensory nerve directly acts on MSCs by binding with FGFR1 to maintain mesenchyme tissue homeostasis.

## FGF signaling depletion in MSCs disturbs mesenchymal stem cell homeostasis

To elucidate how FGF signaling affects incisor growth and tissue homeostasis, we depleted *Fgfr1* gene in the MSCs of *Gli1CreER;Fgfr1fl/fl* mice. The number of *Gli1*+ cells was significantly reduced in *Gli1CreER;Fgfr1fl/fl;Gli1-LacZ* mice one week following TMX induction (Fig. 5a–e). The dynamic turnover of the incisor is driven by MSCs giving rise to TACs, which then differentiate into odontoblasts to generate dentin. This process allows the incisor to maintain continuous growth and mesenchymal tissue homeostasis. To analyze how the loss of FGF signaling might have affected the TACs, we assessed the number of TACs with Ki67 staining. The number of TACs reduced significantly one week after TMX induction in *Gli1CreER;Fgfr1fl/fl* mice (Fig. 5f–j). Since both MSC and TAC numbers had decreased one week after induction, we sought to explore which population was adversely affected first by examining earlier time points. At 3 days after TMX induction, the number of MSCs was already decreased (Supplementary Fig. 6h, i, l), but there was no obvious change in the number of TACs yet in *Fgfr1* mutant mice (Supplementary Fig. 6a–e). This indicated that a decrease in MSCs led to a subsequent reduction in the TAC population. When we detected cellular apoptosis 3 days after TMX induction, there was no obvious apoptosis in control samples but increased TUNEL+ cells were found in the proximal mesenchyme of *Gli1CreER;Fgfr1fl/fl;Gli1-LacZ* mice (Supplementary Fig. 6f, g, m), and this apoptotic activity colocalized with *Gli1*+ cells (Supplementary Fig. 6k).

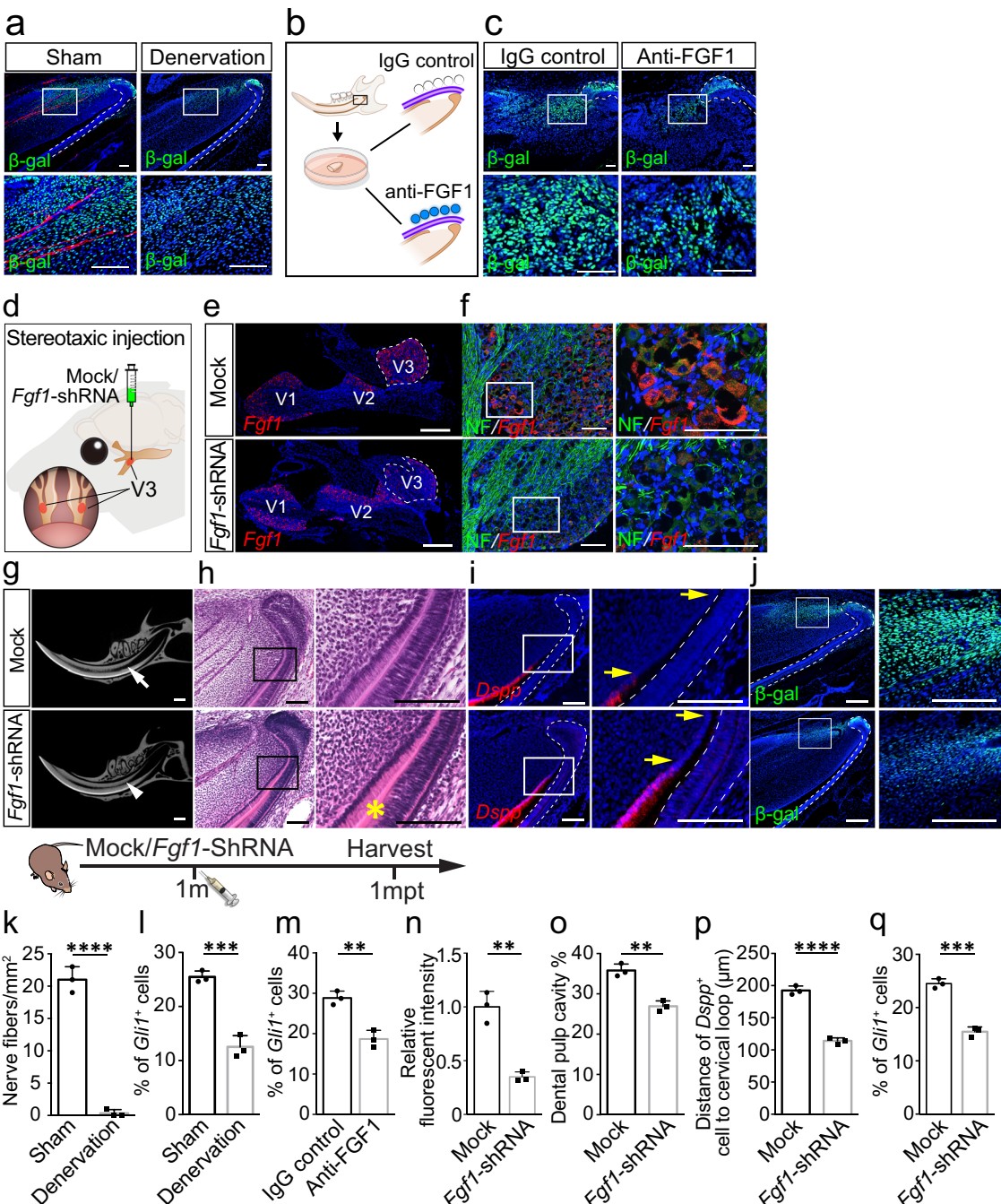

**Fig. 3 | Sensory nerve-derived FGF1 is crucial for MSC maintenance and mesenchymal tissue homeostasis in the incisor. a** Expression of neurofilament and β-gal in sham and denervation groups after one-month denervation. **b** and **c** The proximal ends of *Gli1-LacZ* mouse incisors surrounded by IAN were cultured with IgG or FGF1 antibody-loaded beads. **b** Schematic of the incisor explant culture. **c** *Gli1*+ cells in incisor explant with IAN surrounding the proximal end cultured with IgG-loaded or FGF1 antibody-loaded beads for 3 days. **d** The schematic of stereotaxic injection of *Fgf1*-shRNA. **e** and **f** *Fgf1* decreased in V3 of trigeminal ganglion after *Fgf1*-shRNA injection. **e** Expression of *Fgf1* in trigeminal ganglion with mock shRNA or *Fgf1*-shRNA. V1, V2, and V3 indicate the three branches of the trigeminal nerve. **f** Expression of *Fgf1* and neurofilament in V3 of trigeminal ganglion with mock shRNA or *Fgf1*-shRNA. **g** CT scanning of control and *Fgf1*-shRNA mice one month after injection. White arrow points to the dental pulp cavity; the white arrowhead points to narrowed pulp cavity. **h** Histological analysis of control and *Fgf1*-shRNA mice one month after injection. Asterisk points to abnormal dentin formation. **i** *Dspp* expression in control and *Fgf1*-shRNA mice. Yellow arrows point

to the distance between the bending point of the cervical loop and the initiation of odontoblast differentiation. **j** *Gli1*+ cells stained with β-gal in control and *Fgf1*-shRNA mice. Schematic at the bottom indicates the induction protocol. **k** Quantification of nerve fibers in sham and denervation group. *P* < 0.0001. **l** Quantification of *Gli1*+ cells in sham and denervation group. *P* = 0.0007. **m** Quantification of *Gli1*+ cells in IgG and FGF1 antibody group. *P* = 0.0031. **n** Relative fluorescent intensity of *Fgf1*. *P* = 0.0017. **o** Quantification of dental pulp cavity percentage in control and *Fgf1*-shRNA mice. *P* = 0.0018. **p** Quantification of the distance of *Dspp*+ cells to cervical loop in control and *Fgf1*-shRNA mice. *P* < 0.0001. **q** Quantification of *Gli1*+ cells in control and *Fgf1*-shRNA mice. *P* = 0.0003. For **k**–**q** *n* = 3 and each data point represents one animal, with unpaired Student's *t*-test performed. All data are expressed as the mean ± SD. Source data are provided as a Source Data file. **P* < 0.01, ***P* < 0.001, *****P* < 0.0001. Each experiment was repeated independently three times. White dotted line outlines the cervical loop. Scale bars, **g** 1 mm; **e** 500 μm; others, 100 μm.

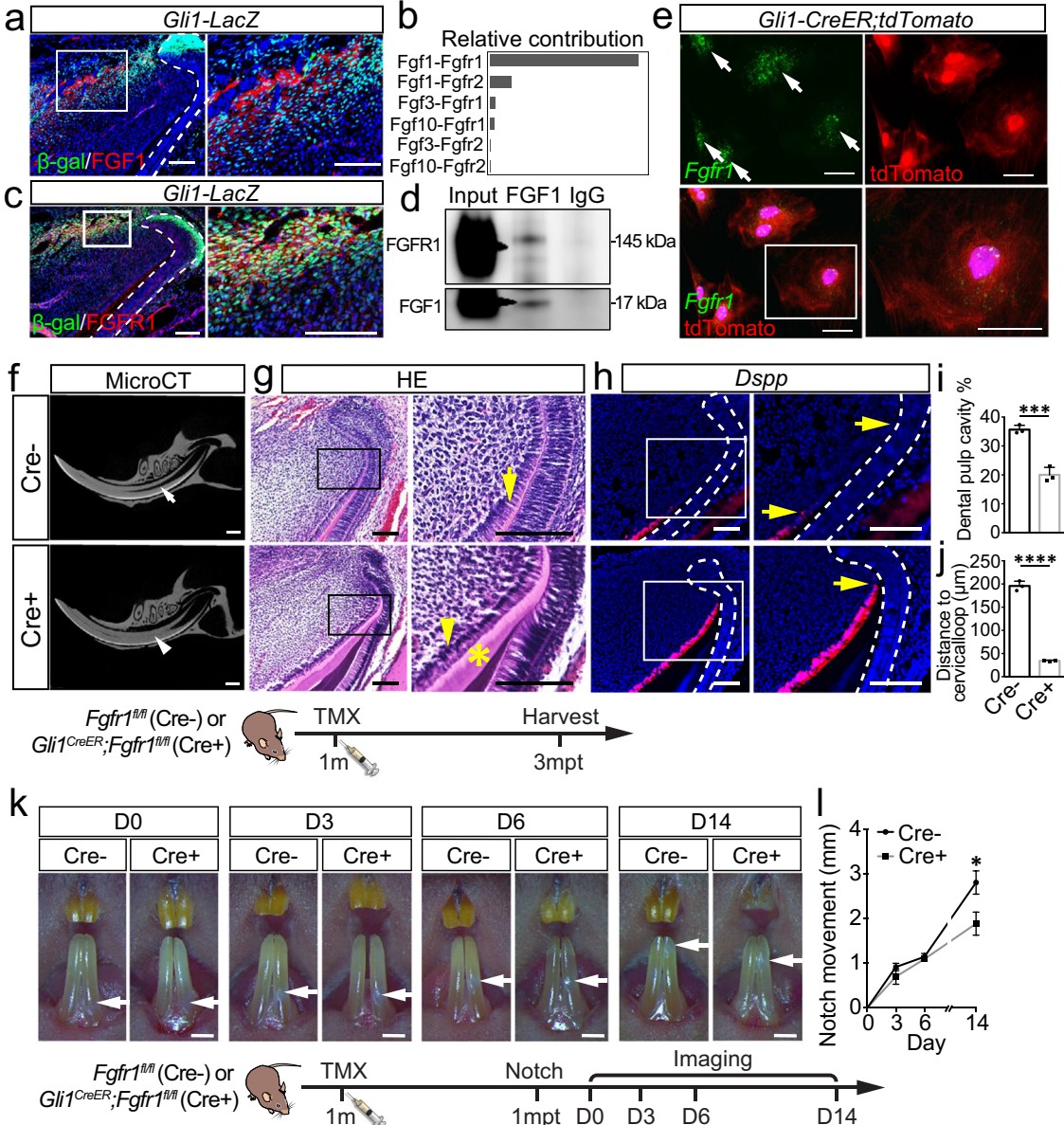

**Fig. 4 | FGF1 secreted from sensory nerves directly acts on MSCs via FGFR1.**
**a** Expression of FGF1 and β-gal in one-month-old *Gli1-LacZ* mice. **b** Relative contributions of ligand-receptor pairs to the overall communication network of FGF signaling pathway; the Fgf1-Fgfr1 pair is the major contributor. **c** Expression of FGFR1 and β-gal in one-month-old *Gli1-LacZ* mice. **d** Immunoprecipitation assay demonstrating the interaction between FGF1 and FGFR1 in the proximal mesenchyme of the incisor. **e** Expression of *Fgfr1* and tdTomato in MSC cultures from *Gli1^CreER^;tdTomato* mice. **f–j** Abnormal dentin formation in *Gli1^CreER^;Fgfr1^fl/fl^* mice three months after tamoxifen induction. **f** CT imaging of the incisors of *Fgfr1^fl/fl^* and *Gli1^CreER^;Fgfr1^fl/fl^* mice. White arrow points to the dental pulp cavity; the white arrowhead points to narrowed pulp cavity. **g** Histological analysis of *Fgfr1^fl/fl^* and *Gli1^CreER^;Fgfr1^fl/fl^* mice. Yellow arrow points to pre-odontoblast; yellow arrowhead points to abnormal pre-odontoblast; asterisk points to abnormal dentin formation. **h** Expression of *Dspp* in control and *Gli1^CreER^;Fgfr1^fl/fl^* mice. Yellow arrow points to the distance between the bending point of the cervical loop and the initiation of odontoblast differentiation. **i** Quantification of dental pulp cavity percentage in control and mutant mice. *P* = 0.0008. **j** Quantification of the distance of *Dspp*⁺ cells to cervical loop in control and mutant mice. *P* < 0.0001. Schematic at the bottom indicates the induction protocol. mpt month post-tamoxifen injection. **k** The growth rate of the incisor was detected with notch movement observed at day (D)3, D6, and D14 in control and *Gli1^CreER^;Fgfr1^fl/fl^* mice. White arrow points to the notch location; the schematic at the bottom indicates the induction protocol. mpt month post-tamoxifen injection. **l** Quantification of notch movement in control and mutant mice. *P* = 0.0123. For **l**, **j** and **l**, *n* = 3 and each data point represent one animal, with unpaired Student's *t*-test performed. All data are expressed as the mean ± SD. Source data are provided as a Source Data file. ***P < 0.001, ****P < 0.0001. Each experiment was repeated independently three times. White dotted line outlines the cervical loop. Cre-: *Fgfr1^fl/fl^*; Cre+: *Gli1^CreER^;Fgfr1^fl/fl^*. Scale bars, **f** 1 mm; **k** 2 mm; **e** 50 μm; others, 100 μm.

This showed that loss of FGF signaling led to apoptosis in MSCs, which contributed to the reduction of MSCs and TACs subsequently. To examine whether the TAC differentiation was altered, we injected EdU 5 days after TMX induction and collected samples after 2 days. Overlap between *Dspp*⁺ odontoblasts and EdU⁺ cells indicated the TACs undergoing odontoblast differentiation. The differentiation of TACs was impaired in *Gli1^CreER^;Fgfr1^fl/fl^* mice (Fig. 5k–o). Meanwhile, we cultured MSCs from *Fgfr1^fl/fl^* and *Gli1^CreER^;Fgfr1^fl/fl^* mice, and analyzed *Dspp* expression in these cells 3 days after mineralization medium (MM) induction for evaluating their odontoblastic differentiation ability. However, the number of *Dspp*⁺ cells was decreased in *Gli1^CreER^;Fgfr1^fl/fl^* MSCs treated with mineralization medium (Supplementary Fig. 6n–r), suggesting that the differentiation capacity of MSCs was impaired following the loss of FGF signaling.

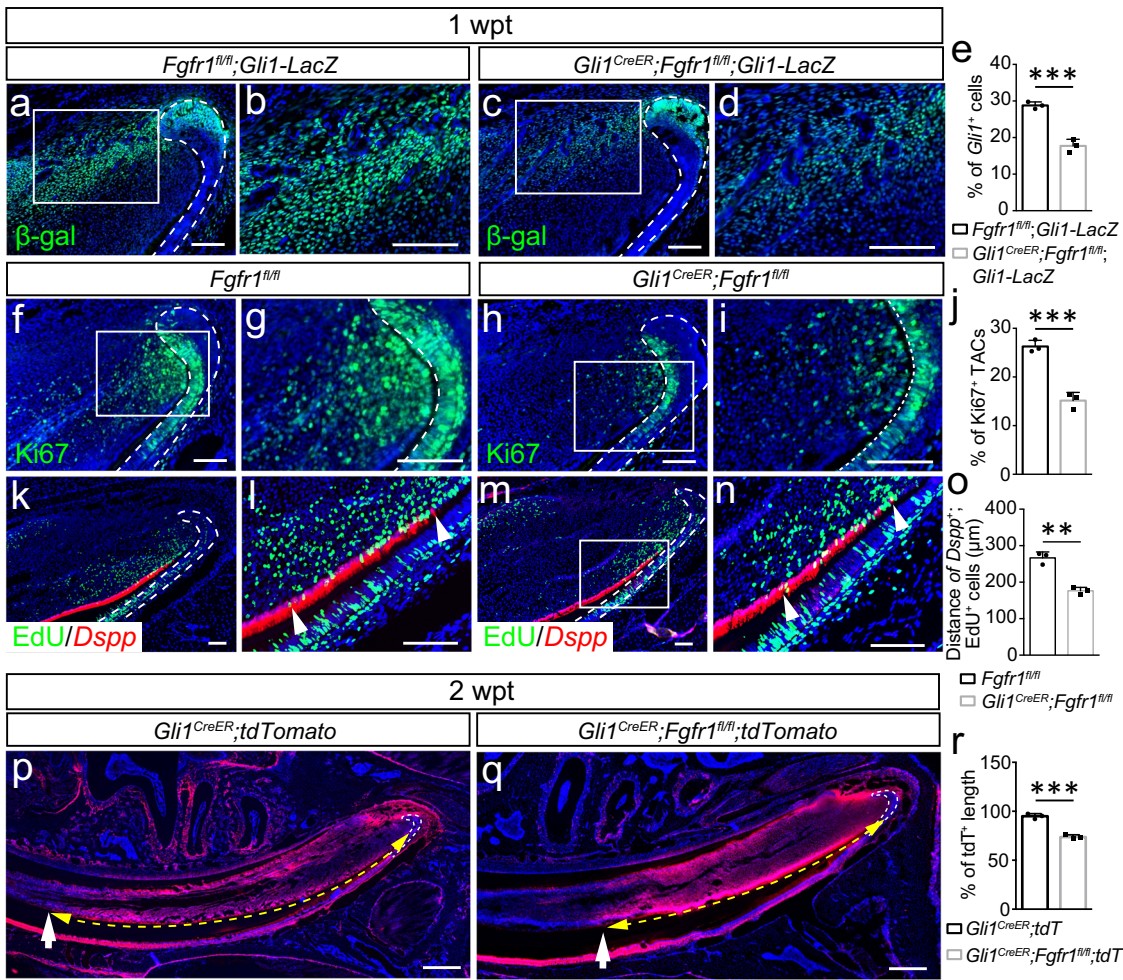

**Fig. 5 | FGF signaling depletion disturbs stem cell homeostasis. a–d** The number of MSCs decreased in *Gli1^CreER;Fgfr1^fl/fl* mice one week after tamoxifen induction. *Gli1^+* cells stained with β-gal in incisors of control and *Gli1^CreER;Fgfr1^fl/fl* mice. **e**, Quantification of the percentage of *Gli1^+* cells in control and mutant mice. *P* = 0.0007. **f–i** TACs detected with Ki67 staining in control and *Gli1^CreER;Fgfr1^fl/fl* mice. **j** Quantification of Ki67^+ TACs in control and mutant mice. *P* = 0.0008. **k–n** The expression of *Dspp* and EdU in control and *Fgfr1^fl/fl* mutant mice. The length of overlap between *Dspp^+* odontoblasts and EdU^+ cells reflects the number of TACs undergoing odontoblastic differentiation. White arrowhead points to overlap between *Dspp^+* odontoblasts and EdU^+ cells. **o** Quantification of overlap between *Dspp^+* odontoblasts and EdU^+ cells in control and mutant mice. *P* = 0.0011. **p** and **q** The migration of *Gli1^+* cells' progeny indicated with tdTomato in *Gli1^CreER;tdTomato* and *Gli1^CreER;Fgfr1^fl/fl;tdTomato* mice. White arrows point to tdTomato^+ cell migration. Yellow dot line with arrowheads point to migration distance. **r** Quantification of the percentage of tdTomato length in control and mutant mice. *P* = 0.0005. For **e**, **o** and **r**, *n* = 3 and each data point represents one animal, with unpaired Student's *t*-test performed. All data are expressed as the mean ± SD. Source data are provided as a Source Data file. **P* < 0.01, ***P* < 0.001. Each experiment was repeated independently three times. White dotted line outlines the cervical loop. Scale bars, **p** and **q** 400 μm; others, 100 μm.

Since both the number and differentiation of TACs decreased in *Gli1^CreER;Fgfr1^fl/fl* mice, we sought to determine how the abnormal dentin and narrowed dental pulp phenotypes emerged. We generated *Gli1^CreER;tdTomato* and *Gli1^CreER;Fgfr1^fl/fl;tdTomato* mice and observed the migration of *Gli1^+* cells' progeny. We measured the length of migration, as indicated by the tdTomato signal, by calculating the ratio between the length of the tdTomato signal and the length of the dental pulp. This ratio decreased two weeks after induction, which suggested that the migration of *Gli1^+* cells' progeny decreased in *Gli1^CreER;Fgfr1^fl/fl* mice (Fig. 5p–r). Furthermore, double labeling with calcein and Alizarin red injection in vivo was used to investigate the dynamics of mineralization (Supplementary Fig. 6s, v). The difference in the lengths of red and green fluorescent signals (Supplementary Fig. 6t, w) indicated odontoblast migration over the course of 5 days in proximal-to-distal direction, which decreased in *Gli1^CreER;Fgfr1^fl/fl* mice (Supplementary Fig. 6t, w, y). The thickness between the red and green fluorescent signals (in Supplementary Fig. 6u, x), which indicated the dentin accumulation within these 5 days, increased in *Gli1^CreER;Fgfr1^fl/fl* mice (Supplementary Fig. 6u, x, z). These findings demonstrated that the abnormal dentin formation was caused by slower

migration and abnormal dentin deposition in *Gli1^CreER;Fgfr1^fl/fl* mice. Collectively, our findings showed that FGF signaling is important for MSC regulation in adult mouse incisors, which influences MSC maintenance, TAC number, and odontoblast differentiation and migration. Disruption of FGF1/FGFR1 interaction in MSCs disturbs mesenchymal tissue homeostasis in the adult mouse incisor.

## FGF signaling activates mTOR/autophagy pathway to sustain MSC

To explore the potential mechanism of FGF signaling in regulating MSC homeostasis, we performed RNAseq of control and *Gli1^CreER;Fgfr1^fl/fl* mice to compare the gene expression profiles in the proximal end of the incisor. We performed hierarchical clustering to confirm that the gene expression profiles of control and *Gli1^CreER;Fgfr1^fl/fl* mice were well separated (Fig. 6a). Of the 3466 differentially expressed genes we identified (>1.5-fold, *P* < 0.05), 2019 were upregulated and 1447 were downregulated (Fig. 6b). Ingenuity pathway analysis (IPA) analysis identified several signaling pathways that were highly enriched (Fig. 6c), which included mitochondrial dysfunction, mTOR signaling,

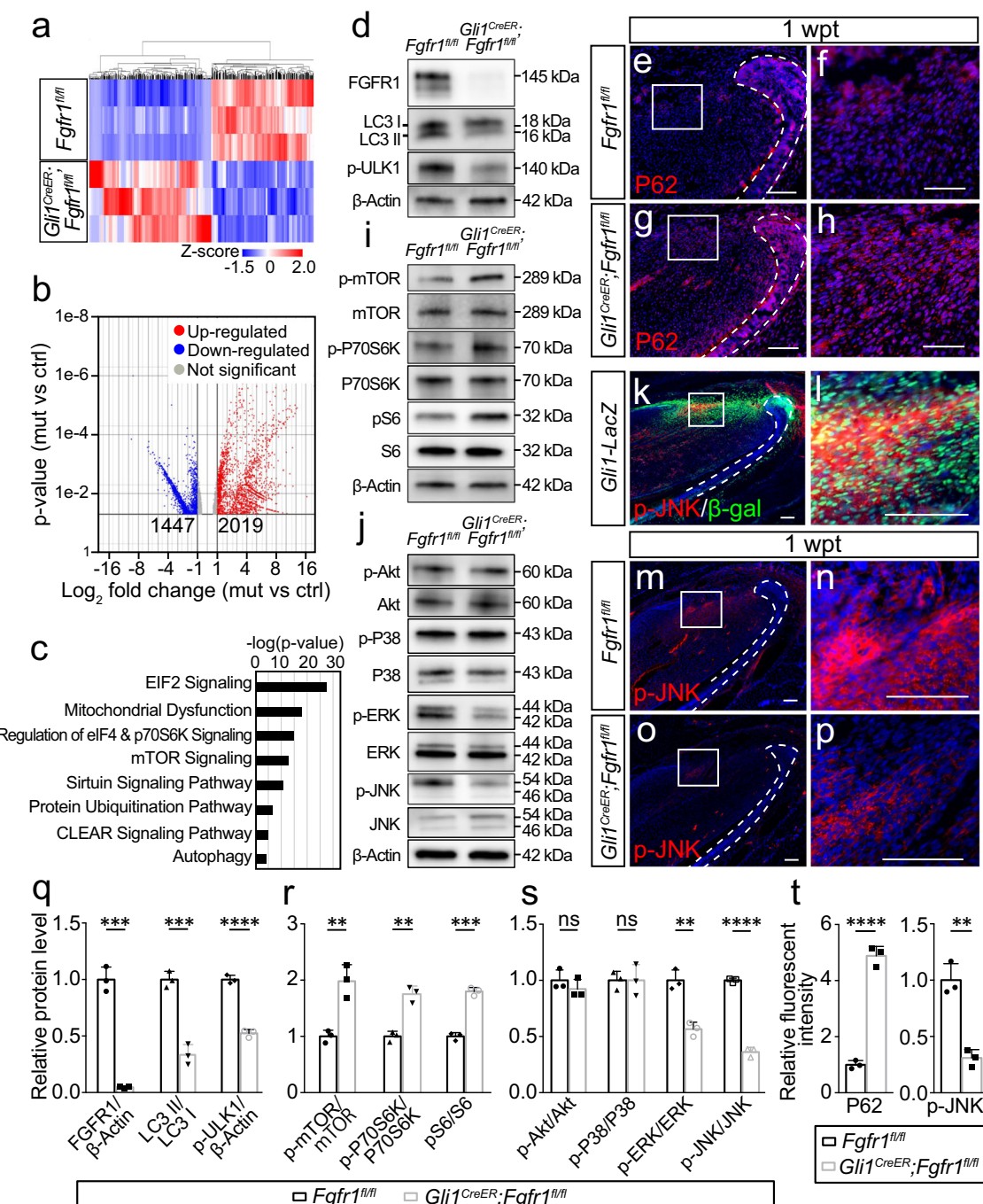

**Fig. 6 | FGF signaling activates mTOR-dependent autophagy in MSCs. a–i** mTOR-dependent autophagy is downregulated in *Gli1^{CreER}*;*Fgfr1^{fl/fl}* mice. **a** Hierarchical clustering of control and *Gli1^{CreER}*;*Fgfr1^{fl/fl}* mice. **b** Volcano plot showing 2019 upregulated genes and 1447 downregulated genes. **c** Significant signaling pathways analyzed with Ingenuity Pathway Analysis (IPA). **d** Expression of FGFR1, LC3 and p-ULK1 in mesenchyme of incisors from control and *Gli1^{CreER}*;*Fgfr1^{fl/fl}* mice. **e–h** Expression of autophagy substrate P62 in control and *Gli1^{CreER}*;*Fgfr1^{fl/fl}* mice. **i** Expression of p-mTOR and its downstream effectors p-P70S6K and pS6 in control and mutant mice. **j–p** FGF/p-JNK signaling regulates mTOR-dependent autophagy activation in MSCs. **j** Western blot of p-AKT, AKT, p-P38, P38, p-ERK, ERK, p-JNK, and JNK in proximal mesenchyme from control and mutant mice. **k** and **l** Expression of p-JNK and β-gal in the proximal mesenchyme of incisors from *Gli1-*

*LacZ* mice. **m–p** Expression of p-JNK in control and *Gli1^{CreER}*;*Fgfr1^{fl/fl}* mice. **q** Relative protein level in (**d**). FGFR1/β-Actin: *P* = 0.0001; LC3II/LC3I: *P* = 0.0006; p-ULK1/β-Actin: P < 0.0001. **r** Relative protein level in (**i**). p-mTOR/mTOR: P = 0.0056; p-P70S6K/P70S6K: *P* = 0.0016; pS6/S6: *P* = 0.0001. **s** Relative protein level in (**j**). p-Akt/Akt: *P* = 0.3485; p-P38/P38: *P* = 0.9988; p-ERK/ERK: *P* = 0.0025; p-JNK/JNK: *P* < 0.0001. **t** Relative fluorescent intensity of P62 and p-JNK. P62: *P* < 0.0001; p-JNK: *P* = 0.0019. For **q–t**, *n* = 3 and each data point represent one biological replicate, with unpaired Student's *t*-test performed. All data are expressed as the mean ± SD. Source data are provided as a Source Data file. **P < 0.01, ***P < 0.001, ****P < 0.0001. Each experiment was repeated independently three times. White dotted line outlines the cervical loop. Scale bars, 100 μm.

clear signaling, and autophagy. These signaling pathways are closely related to the mTOR/autophagy axis. mTOR signaling participates in autophagy regulation, and autophagy-related genes were also found to be downregulated in the gene list. For these

reasons, we focused on mTOR/autophagy signaling in our subsequent investigation.

Based on recent studies, autophagy plays important roles in the homeostasis of different stem cell populations, such as neural,

hematopoietic, muscle, cancer, and induced pluripotent stem cells[28]. To explore whether FGF signaling regulates MSC homeostasis by modulation of autophagy in the adult mouse incisor, we first detected autophagy-related protein expression in control and *Gli1^CreER;Fgfr1^fl/fl* mice. We extracted proteins from the proximal mesenchyme of the incisor and found that expression of FGFR1 was decreased in *Gli1^CreER;Fgfr1^fl/fl* mice. The autophagosomal protein LC3 is a frequently used marker by which the autophagy process can be followed. Protein levels of LC3II and the autophagy initiation protein p-ULK1 decreased in the proximal region of the incisor of *Gli1^CreER;Fgfr1^fl/fl* mice (Fig. 6d, q). Then, we detected the autophagy substrate P62, which marks damaged organelles for degradation by selective autophagy. In control mice, no obvious P62 signal was detected in the proximal mesenchyme (Fig. 6e, f), whereas accumulation of P62 was found in the proximal mesenchyme of *Gli1^CreER;Fgfr1^fl/fl* mice (Fig. 6g, h). Furthermore, we detected mTOR signaling that negatively regulated autophagy. The expression of p-mTOR increased as well as its downstream effector p-P70S6K and pS6 (Fig. 6i, r). These findings suggested that mTOR-dependent autophagy was downregulated in *Gli1^CreER;Fgfr1^fl/fl* mice.

FGF signaling activates several signaling pathways, such as AKT, ERK, JNK and others, which indicated a potential role in the regulation of autophagy. To determine the downstream signaling pathway of which FGF/FGFR1 signal targets, we collected the proximal incisor mesenchyme of control and *Gli1^CreER;Fgfr1^fl/fl* mice and protein samples were extracted for Western blot analysis. The expression of p-ERK and p-JNK were decreased in *Gli1^CreER;Fgfr1^fl/fl* mice, while no obvious change in p-AKT or p-P38 between control and *Fgfr1* mutant mice was observed (Fig. 6j, s). Analyzing *Gli1-LacZ* mice, we found that p-JNK was expressed in *Gli1*+ cells in the proximal incisor mesenchyme (Fig. 6k, l), while p-ERK was undetectable in *Gli1*+ cells (Supplementary Fig. 7a–c). It indicated that JNK may be the downstream target of FGF signaling. The expression of p-JNK was decreased in *Gli1^CreER;Fgfr1^fl/fl* mice (Fig. 6m–p, t). These results demonstrate that FGF/p-JNK signaling regulates mTOR-dependent autophagy activation in MSCs.

## Activation of autophagy rescues MSCs in *Gli1^CreER;Fgfr1^fl/fl* mice

To confirm the role of autophagy, we re-activated autophagy in *Gli1^CreER;Fgfr1^fl/fl* mice with rapamycin. Rapamycin is a US Food and Drug Administration (FDA)-approved drug known to activate autophagy[31,37]. We found accumulation of the autophagy substrate P62 in the proximal incisor mesenchyme in *Gli1^CreER;Fgfr1^fl/fl* mice, while its expression was decreased with rapamycin treatment (Supplementary Fig. 7d–i). This suggested that autophagy was re-activated in *Gli1^CreER;Fgfr1^fl/fl* mice with rapamycin treatment. Three months after TMX induction, the abnormal dentin formation, and narrowed dental pulp cavity phenotypes were rescued in rapamycin-treated *Gli1^CreER;Fgfr1^fl/fl* mice (Fig. 7a, d). Histological analysis revealed that the abnormal dentin in the proximal area of the incisor was reversed by rapamycin treatment (Fig. 7b). *Dspp* expression was found closer to the proximal end of the incisor in *Gli1^CreER;Fgfr1^fl/fl* mice, but was restored to the normal pattern in *Fgfr1* mutant mice with rapamycin treatment (Fig. 7c, e). These findings collectively suggested that the re-activation of autophagy rescues mesenchymal disorder. The loss of *Gli1*+ cells in *Gli1^CreER;Fgfr1^fl/fl;Gli1-LacZ* mice was rescued by rapamycin treatment (Fig. 7f, g), which suggests that autophagy re-activation benefits the maintenance of MSCs. Since we previously found that a decrease in MSCs leads to a reduction in the number of TACs, we detected the TACs with Ki67 staining to see whether the TACs were rescued after MSCs were restored with rapamycin treatment. The decreased number of TACs in the *Gli1^CreER;Fgfr1^fl/fl* mice was also restored by rapamycin treatment (Fig. 7h, i). Then we detected apoptosis and found that TUNEL+ cells were decreased with rapamycin treatment (Supplementary Fig. 8a). These results suggested that autophagy is essential to MSC maintenance and aids tissue homeostasis in the adult mouse incisor.

There is an association between mitochondrial dysfunction and increased reactive oxygen species (ROS)[38], which is an important trigger of apoptosis. Significantly, mitochondrial dysfunction was detected in the incisor of *Gli1^CreER;Fgfr1^fl/fl* mice. To analyze whether FGFR1-regulated autophagy is responsible for proper mitochondrial function and MSCs survival, we analyzed ROS expression in MSCs from control and *Gli1^CreER;Fgfr1^fl/fl* mouse incisors treated with FGF1 or rapamycin. ROS was barely detectable in control MSCs after FGF1 stimulation, while there were increased ROS+ MSCs from *Gli1^CreER;Fgfr1^fl/fl* mice (Supplementary Fig. 8f, g). This indicated that loss of FGF signaling in MSCs led to ROS accumulation. To further investigate if autophagy activation is FGF signaling dependent in MSCs of adult mouse incisor, we performed a study to show that there were LC3 puncta in the cytoplasm of MSCs after FGF1 stimulation, but LC3+ cells were decreased in MSCs from *Gli1^CreER;Fgfr1^fl/fl* mice (Supplementary Fig. 8c, d). It suggested that autophagy could not be activated by FGF1 treatment after the depletion of FGFR1 in MSCs. However, autophagy was activated with rapamycin treatment in MSCs from both control and *Gli1^CreER;Fgfr1^fl/fl* mice (Supplementary Fig. 8c, e). Furthermore, ROS were cleared in *Fgfr1* mutant MSCs with rapamycin treatment (Supplementary Fig. 8f, h). These results demonstrated that autophagy is downstream of FGF signaling. The clearance of ROS through autophagy is crucial to protect MSCs from undergoing apoptosis.

To check whether reduced autophagy affects cell migration, we observed MSC migration with and without FGF1 or rapamycin treatment. MSCs from *Gli1^CreER;Fgfr1^fl/fl* mice showed slower migration than control at 12 and 24 h (Supplementary Fig. 8i). Loss of FGF signaling impaired cell migration, which was consistent with the slow migration of odontogenic progenitors in *Gli1^CreER;Fgfr1^fl/fl* mice. The cell migration was enhanced by FGF1 treatment, but was still slower in *Fgfr1* mutant mice in comparison to the controls (Supplementary Fig. 8j). Rapamycin promoted the migration in both control and *Fgfr1* mutant cells (Supplementary Fig. 8k). These findings suggested that autophagy is downstream of FGF signaling in modulating cell migration, and that re-activation of autophagy can restore the proper odontogenic progenitor migration. Autophagy serves an important role as a downstream target of FGF1/FGFR1 signaling to regulate MSC-mediated tissue homeostasis via conferring resistance to apoptosis and sustaining MSCs. Activation of autophagy benefits the maintenance and apoptosis resistance of MSCs, as well as cell migration and tissue homeostasis in the adult mouse incisor.

## Discussion

In this study, we have uncovered direct interaction between sensory nerves and MSCs in the adult mouse incisor model. FGF1, a ligand secreted by sensory nerves, directly regulates MSCs in the incisor by binding to FGFR1 and activates the FGF/p-JNK/mTOR/autophagy axis to regulate MSCs in maintaining tissue homeostasis (Fig. 8). Loss of FGF1/FGFR1-mediated nerve–MSC interaction leads to a compromised MSCs in adult mouse incisor. Sensory nerves mediate the retention, survival, and differentiation of MSCs in adult mouse incisors, serving as a stem cell niche that controls MSCs and mesenchymal tissue homeostasis.

The mouse incisor is a highly innervated organ that grows continually owing to the consistent presence of stem cells. In this study, we have examined sensory, sympathetic, and parasympathetic nerves in the adult mouse incisor. Unlike bone, which is innervated with both sensory and sympathetic nerves[17], the majority of nerves in the incisor are CGRP+ sensory nerves. Sensory nerve predominates in innervating the mouse incisor, which confers the mouse incisor as an ideal model for the study of the relationship between sensory nerves and MSCs. We show that sensory nerves regulate the MSCs maintenance and mesenchymal tissue homeostasis of the incisor. Loss of sensory nerves leads to decreased number of MSCs and osteogenic progenitor cells in bone and osteogenesis disruption[22,25]. Taken together, these studies

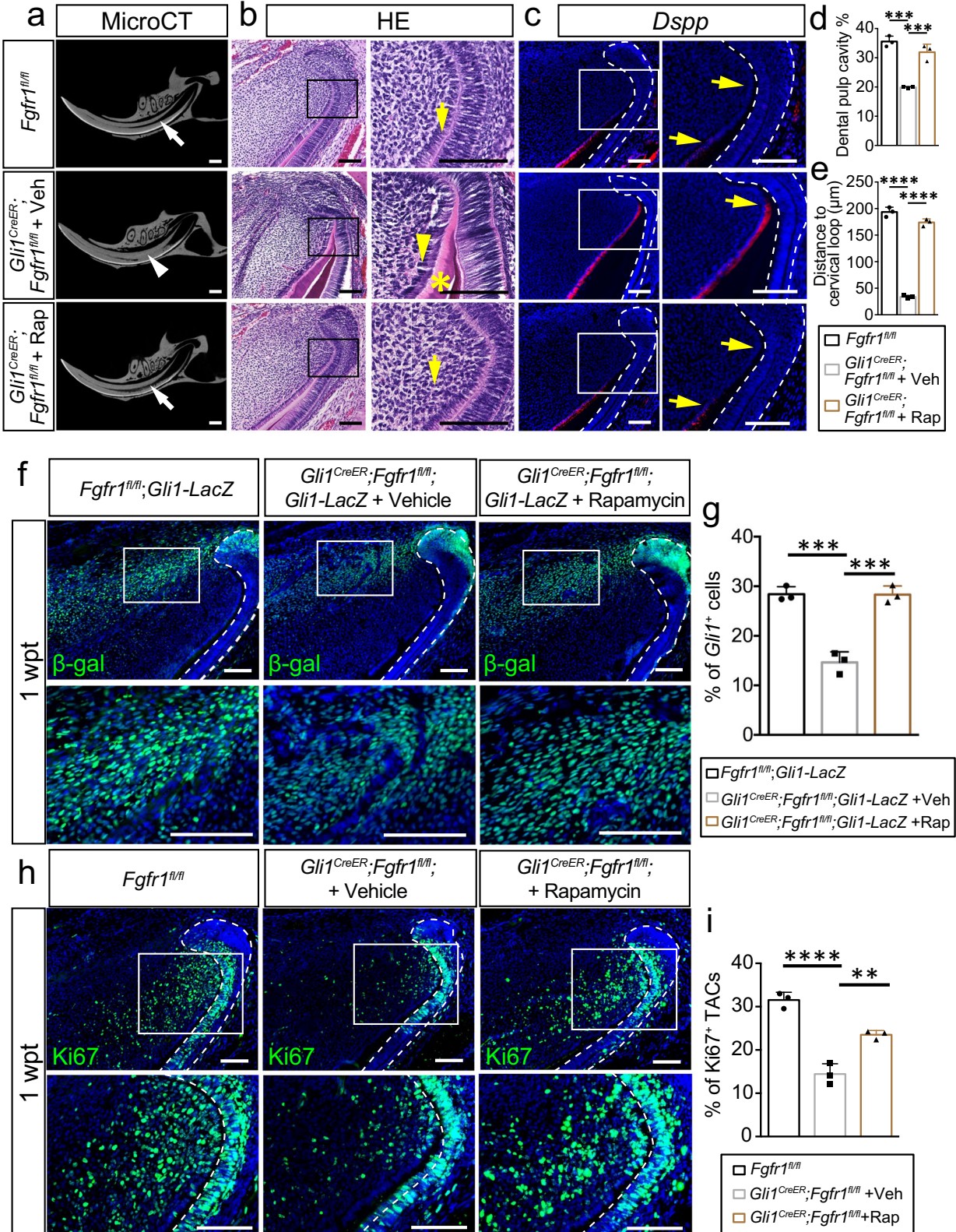

uncover the essential role of sensory nerves in maintaining MSCs in different tissues.

We have found that FGF signaling mediates the crucial interaction between sensory nerves and proximal mesenchymal cells in adult mouse incisors. Recent study has shown that sensory nerves regulate mesenchymal progenitor cells and cranial bone formation through BMP/TGF-β signaling[25]. Signaling pathways including FGF, TGF-β,

PDGF, VEGF, and WNT are known to be involved in the sensory nerve–bone crosstalk in aberrant osteochondral differentiation after soft tissue trauma[39]. Schwann cell precursors participate in skeletal formation[40], and secrete factors, such as PDGF-AA and oncostatin M, to involve in the mandibular repair[15] and digit regeneration[14]. In our study, PDGF signaling may also be involved in regulating the interaction between sensory nerve and proximal mesenchymal cells in adult

**Fig. 7 | Activation of autophagy sustains MSCs. a–e** Abnormal dentin deposition seen in *Gli1^CreER^;Fgfr1^fl/fl^* mice can be rescued by rapamycin treatment for 3 months. **a** CT scanning of control, *Gli1^CreER^;Fgfr1^fl/fl^* mice treated with vehicle (*Gli1^CreER^;Fgfr1^fl/fl^* + Veh) and *Gli1^CreER^;Fgfr1^fl/fl^* mice treated with rapamycin (*Gli1^CreER^;Fgfr1^fl/fl^* + Rap). White arrow points to the dental pulp cavity; white arrowhead points to narrowed pulp cavity. **b** Histological analysis of these three groups. Yellow arrow points to normal pre-odontoblast; yellow arrowhead points to abnormal pre-odontoblast; asterisk points to abnormal dentin formation. **c** Expression of *Dspp* in these three groups. Yellow arrow points to the distance between the bending point of the cervical loop and the initiation of odontoblast differentiation. **d** Quantification of dental pulp cavity percentage in these three groups. Control vs *Gli1^CreER^;Fgfr1^fl/fl^* + Veh: *P* = 0.0001; *Gli1^CreER^;Fgfr1^fl/fl^* + Veh vs. *Gli1^CreER^;Fgfr1^fl/fl^* + Rap: *P* = 0.0006. **e** Quantification of the distance of *Dspp^+^* cells to cervical loop. Control versus

*Gli1^CreER^;Fgfr1^fl/fl^* + Veh: *P* < 0.0001; *Gli1^CreER^;Fgfr1^fl/fl^* + Veh versus *Gli1^CreER^;Fgfr1^fl/fl^* + Rap: *P* < 0.0001. **f** Re-activation of autophagy benefits the retention of MSCs. *Gli1^+^* cells labeled with β-gal in control, *Gli1^CreER^;Fgfr1^fl/fl^* mice treated with vehicle or rapamycin. **g** Quantification of the percentage of *Gli1^+^* cells. Control versus *Gli1^CreER^;Fgfr1^fl/fl^* + Veh: *P* = 0.0002; *Gli1^CreER^;Fgfr1^fl/fl^* + Veh versus *Gli1^CreER^;Fgfr1^fl/fl^* + Rap: *P* = 0.0002. **h** TACs detected with Ki67 staining in control, *Gli1^CreER^;Fgfr1^fl/fl^* mice treated with vehicle or rapamycin. **i** Quantification of Ki67^+^ TAC cells. Control versus *Gli1^CreER^;Fgfr1^fl/fl^* + Veh: *P* < 0.0001; *Gli1^CreER^;Fgfr1^fl/fl^* + Veh versus *Gli1^CreER^;Fgfr1^fl/fl^* + Rap: *P* = 0.0021. For **d**, **e**, **g**, and **I**, *n* = 3 biologically independent samples, each data point represents one animal, with unpaired one-way ANOVA analysis. All data are expressed as the mean ± SD. Source data are provided as a Source Data file. \*\**P* < 0.01, \*\*\**P* < 0.001, \*\*\*\**P* < 0.0001. Each experiment was repeated independently three times. White dotted line outlines the cervical loop. Scale bars, 100 μm.

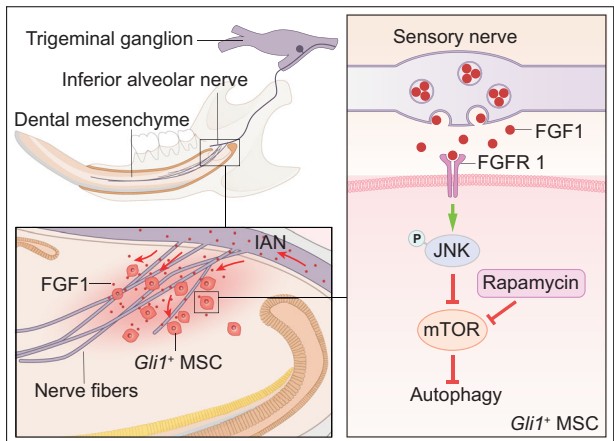

**Fig. 8 | Schematic of sensory nerve niche regulating MSC homeostasis in adult mouse incisor.** Sensory nerve predominates in innervating the mouse incisor, which confers the mouse incisor as an ideal model for the study of the relationship between sensory nerves and MSCs. FGF1, a ligand secreted by sensory nerves, is enriched in the proximal mesenchymal end of the incisor and surrounding *Gli1^+^* cell. FGF1 directly regulates MSCs in the incisor by binding to FGFR1 specifically and activates the FGF/p-JNK/mTOR/autophagy axis to regulate MSCs in maintaining tissue homeostasis.

mouse incisors based on our bioinformatic analysis. However, this will require further investigation. These findings suggest that sensory nerves serve as a niche and interact with various components in the incisor through ligand–receptor interactions.

Sensory nerves release various peptide and nonpeptide neurotransmitters, such as Substance P (SP), calcitonin gene-related peptide (CGRP), neurokinin-A (NK-A), secretoneurin (SN), somatostatin (SOM), and vasoactive intestinal peptide (VIP)[41]. Whether sensory nerves could directly regulate MSCs by releasing other ligands is largely unknown. In our study, we have explored the communication between sensory nerves and the incisor. FGF signaling plays a significant role, with sensory neurons supplying FGF1 and MSCs receiving it through the specific receptor FGFR1. FGFs are regarded as paracrine factors which are essential for organogenesis and tissue homeostasis. The FGF1 is secreted independently through the endoplasmic reticulum–Golgi secretory pathway, but its role is yet to be fully explored[33]. In recent years, the injection of recombinant FGF1 has been used to lower glucose levels in hyperglycemic diabetic mice[42]. The central injection of recombinant FGF1 can restore glucose-sensing neurons and synaptic function, which eventually remedy the hypothalamic state to achieve long-lasting glucose-lowering effects[43].

Here we show that FGF1 derived from the sensory nerve is important for MSC homeostasis in the mouse incisor model. Previous studies have shown that FGF signaling is essential for the maintenance of stem cells. Stimulating bone marrow cells with FGF1 alone results in

long-term repopulation of HSCs in vitro[44]. OCT4 can bind to the *FGF2* gene to sustain self-renewal in human embryonic stem cells[45]. FGF and IGF coordinately regulate the stem cell niche of human pluripotent stem cells and inhibiting the FGF signaling pathway cause them to differentiate[46]. Our study has also revealed local FGF signaling in the incisor, which originated from TACs and odontoblasts. Previous studies have shown that FGF ligands in the incisor, such as FGF3 and FGF10, regulate incisor epithelium through FGFR1 and FGFR2[34,35]. CellChat analysis of the dental pulp FGF signaling network has shown that TACs are the major source of FGF signaling in the dental mesenchyme and that this signal predominantly targets dental follicle and pulp cells. This suggests that, along with FGF1 derived from the sensory nerve, other FGF ligands in the incisor synergistically regulate mesenchymal cells.

FGF1 exerts its function in an autocrine/paracrine manner, and its binding with heparan sulfate proteoglycans serves to prevent them from entering the circulation[33,43]. Understanding the FGF–FGFR interaction is key to revealing how FGF1 secreted from sensory neurons regulates the fate of MSCs. Based on our comprehensive analysis, FGF1 secreted from sensory nerves binds directly with the specific receptor FGFR1 on MSCs, activating FGF signaling to maintain mesenchymal tissue homeostasis. Furthermore, we show that mTOR/autophagy signaling is affected following the loss of FGF signaling. Previous study shows that the deficiency of *Etv* family genes increase mTOR signaling to regulate organelle removal in lens maturation, and augmented mTOR phosphorylation is caused by the negative regulator *Tsc2*[47]. The mechanistic target of rapamycin complex1 (mTORC1) can be activated by the small GTPase Rheb. The mTORC1 can regulate protein synthesis, lipid and glucose metabolism, and autophagy-related regulation of protein turnover[48,49]. These findings illustrate the activation of mTORC1 after the loss of FGF signaling. However, how FGF/p-JNK regulates mTORC1 still needs to be explored in the future.

Autophagy is the main "quality control" pathway in cells, and as such, plays an important role in regulating stem cells, as has been investigated in different tissues[28,32]. Deletion of FIP200 in HSCs leads to deficient autophagy and a decreased number of liver embryonic HSCs[32]. These HSCs have increased accumulation of mitochondria and ROS, as well as increased DNA damage, after deleting the autophagy gene *Atg7* in the hematopoietic system[30]. We show that blocking FGF1/FGFR1 signaling leads to impaired autophagy as well as a decreased population of MSCs in *Gli1^CreER^;Fgfr1^fl/fl^* mice. Recently, autophagy has been used as a tool to modulate stem cell fate and lifespan. A number of well-established approaches to modulate autophagy, such as rapamycin/spermidine treatment, caloric restriction, and a low-protein diet, have been used to maintain the function of HSCs and muscle satellite cells[28]. Here we show that autophagy is crucial to preserve adult MSCs by reducing oxidative damage and apoptosis in the adult mouse incisor. Re-activated autophagy with rapamycin treatment restores the decreased number

of MSCs and TACs and decreases apoptosis in adult incisor mesenchyme of *Gli1-CreER;Fgfr1^{fl/fl}* mice. Adult MSC function and mesenchymal tissue homeostasis can be modulated by targeting autophagy, which provides a potential therapeutic target for tissue regeneration.

In summary, our study demonstrates that sensory nerves can directly regulate the MSC population through secreting FGF1, which acts on MSCs by binding with FGFR1. This highlights the important role of sensory nerves in regulating stem cells to maintain tissue homeostasis and suggests a new avenue of research in which the ligands secreted from sensory nerves and their interactions with stem cells in different tissues could be explored. In addition, the sensory nerve serves as an essential stem cell niche that modulates MSC homeostasis through the FGF/p-JNK/mTOR/autophagy axis. Re-establishing autophagy with rapamycin sustains MSCs and restores mesenchymal tissue homeostasis. Collectively, our findings contribute to understanding the mechanism by which sensory nerves regulate MSCs to achieve proper tissue homeostasis.

# Methods

## Animals

*Advillin^{CreER}* (JAX#032027)[50], *ROSA-DTA* (JAX#009669)[51], *tdTomato* (JAX#007905)[52], *Gli1^{CreER}* (JAX#007913)[53], *Gli1-LacZ* (JAX# 008211)[54], *K14^{rtTA}* (JAX#007678)[55], *tetO^{Cre}* (JAX#006234)[56], C57BL/6J (JAX#000664), *Fgfr1^{fl/fl}* (JAX #007671)[57], and *Fgfr2^{fl/fl}* (from Dr. Philippe Soriano)[58] mouse lines were used. All these mice were housed in pathogen-free conditions and analyzed with a mixed background. DNA was extracted from the tail, and genotypes were identified through PCR analysis. All mouse studies were conducted with protocols approved by the Department of Animal Resources and the Institutional Animal Care and Use Committee (IACUC) of the University of Southern California (Protocols 9320 and 11765).

## Tamoxifen, doxycycline and rapamycin administration

Tamoxifen (Sigma, T5648) was dissolved in corn oil (Sigma, C8267) at 20 mg/ml. Control, Advillin^{CreER};DTA, Advillin^{CreER};tdTomato, Gli1^{CreER};Fgfr1^{fl/fl}, Gli1^{CreER};Fgfr1^{fl/fl};tdTomato, Gli1^{CreER};Fgfr1^{fl/fl};Gli1-LacZ, and Gli1^{CreER};Fgfr2^{fl/fl} mice were injected intraperitoneally (i.p.) at a dosage of 1.5 mg/10 g body weight at one month of age. Control and K14^{rtTA};tetO^{Cre};Fgfr1^{fl/fl} mice were fed with a doxycycline rodent diet (ENVIGO, TD.08541) daily starting from one month of age. Rapamycin (LC Laboratories) was dissolved in ethanol to 500 mg/ml, then further diluted in 5% PEG-400/5% Tween-80. Control mice were i.p. injected with vehicle solution (5% PEG-400/5% Tween-20), and Gli1^{CreER};Fgfr1^{fl/fl} mice were i.p. injected daily with vehicle solution or rapamycin at a final dosage of 8.0 mg/kg.

## Tissue clearing and staining

Incisors were collected from mice and fixed with 4% paraformaldehyde. The incisors were dissected and transparentized with tissue clearing reagent (TCI, T3741) following the manufacturer's protocol. The incisor was incubated with a neurofilament antibody for 1 day (1:100, Abcam, ab4680) at 4 °C overnight, and an Alexa-conjugated secondary antibody (1:200, Invitrogen) was used to detect signals. Images were captured with a confocal microscope (Leica, Stellaris confocal).

## MicroCT analysis

Incisors were collected from mice and fixed with 4% paraformaldehyde after soft tissue was removed. MicroCT analysis was performed using a SCANCO µCT50 (Scanco V1.28) device at the University of Southern California Molecular Imaging Center. The scanning process was performed at 90 kVp and 78 µA, with a resolution of 10 µm. Visualization and three-dimensional reconstruction were performed using AVIZO 9.1.0 (Visualization Sciences Group).

## EdU incorporation and staining

EdU was dissolved in PBS to 10 mg/ml. Control and *Gli1^{CreER};Fgfr1^{fl/fl}* mice were injected with EdU (50 µg/g body weight) intraperitoneally for 48 h before being euthanized for TAC differentiation analysis. The mandibles were dissected, fixed, and decalcified. We prepared sagittal cryosections of the incisor, and EdU staining was performed according to the manufacturer's instructions using Click-iT™ plus EdU cell proliferation kit (Thermo Fisher Scientific, C10637).

## Histological analysis and immunofluorescence

Mouse mandibles were collected and fixed in 4% paraformaldehyde (PFA) overnight, then decalcified with 10% EDTA in PBS for 2–4 weeks according to the age of the mice. For Hematoxylin and Eosin (H&E) staining, the samples were dehydrated in an ethanol and xylene series and embedded in paraffin, after which samples were cut into 5 µm sections using a microtome (Leica, RM2235 ccwUS). H&E staining was performed using standard protocols. For immunofluorescence (IF) analysis, the decalcified mandibles were dehydrated in serial sucrose solutions and embedded in an OCT compound (Tissue-Tek, Sakura), and samples were cut into 8 µm cryosections using a cryostat (Leica CM1850). For section IF staining, the cryosections were soaked in blocking solution (PerkinElmer, FP1012) for 1 h, then incubated with primary antibodies at 4 °C overnight. For cell IF staining, cells were fixed with formalin, permeabilized with 0.2% Triton X-100, blocked with blocking buffer for 1 h, and incubated with primary antibody at 4 °C overnight. Alexa-conjugated secondary antibody (1:200, Invitrogen) was used to detect signals, followed by counterstaining with DAPI (Invitrogen, 62248). Images were captured with a Keyence microscope (Carl Zeiss).

The primary antibodies used were the following: neurofilament (1:100, Abcam, ab4680), βIIITubulin (TUJ1) (1:100, Abcam, ab18207), CGRP (1:100, Abcam, ab36001), Tyrosine Hydroxylase (TH) (1:100, Millipore, AB152), Choline Acetyltransferase (ChAT) (1:100, Abcam, ab178850), FGF1 (1:100, Abcam, ab169748), Advillin (P92) (1:100, Abcam, ab72210), β galactosidase(β-gal) (1:100, Abcam, ab9361), S100 (1:100, Abcam, ab52642), FGFR1 (1:100, Sigma, HPA056402), p-ERK (1:100, Cell Signaling, 4370), p-JNK (1:100, Cell Signaling, 9255), P62 (1:100, MBL, PM045), Ki67 (1:100, Abcam, ab15580), and LC3 (1:100, Abcam, ab48394).

## In situ hybridization

Staining was performed on cryosections according to the manufacturer's instructions using RNAscope 2.5 HD Reagent Kit-RED assay (Advanced Cell Diagnostics, 322350) or RNAscope Multiplex Fluorescent v2 kit (Advanced Cell Diagnostics, 323100). All probes used in this study were synthesized by Advanced Cell Diagnostics: Probe-Mm-*Fgf1* (466661), Probe-Mm-*Fgfr1* (454941), Probe-Mm-*Fgfr2* (443501), Probe-Mm-*Fgfr3* (440771), Probe-Mm-*Fgfr4* (443511), Probe-Mm-*Dspp* (448301), and Probe-Mm-*Gfra2*-C2 (441481-C2).

## Nuclei isolation and single nuclei RNA-sequencing (snRNA-Seq)

Mice were euthanized by $CO_2$ inhalation and then decapitated. Bilateral trigeminal ganglion tissues were dissected. Nuclei isolation was performed as previously described with minor modifications[59]. Briefly, samples were dounce homogenized with five strokes of a loose pestle and five strokes of a tight pestle in ice-cold detergent lysis buffer (0.1% Triton-X, 0.32 M sucrose, 10 mM HEPES, 5 mM $CaCl_2$, 3 mM MgAc, 0.1 mM EDTA, and 1 mM DTT in nuclease-free water, pH 8.0). The lysate was centrifuged at 3200×*g* for 10 min at 4 °C and the pellet was resuspended with 3 ml low sucrose buffer (0.32 M sucrose, 10 mM HEPES, 5 mM $CaCl_2$, 3 mM MgAc, 0.1 mM EDTA, and 1 mM DTT in nuclease-free water, pH 8.0). The nuclei were isolated and purified by centrifugation in the sucrose density gradient at 3200×*g* for 20 min at 4 °C, and then resuspended with resuspension solution (0.4 mg/ml BSA, 0.2 U/µl RNase inhibitor in DPBS). Approximately 20,000 nuclei

were loaded into a 10X Chromium system with the targeted recovery of 8679 nuclei to be barcoded for snRNA-seq using a Single Cell 3′ Library Kit v3.1 (PN-1000269, 10x genomics). Sequencing was performed on the Illumina Novaseq System. Raw read counts were analyzed using the Seurat R package.

## Notch labeling and incisor injury

Control, *Advillin^CreER;Rosa-DTA*, and *Gli1^CreER;Fgfr1^fl/fl* mice were anesthetized one month after TMX induction. We used a carbide bur (Brasseler USA, 018554U0) to make a notch on enamel above the gumline, and measured the growth length at different time points. For incisor injury, we cut the incisor along the gingiva papilla.

## snRNA-Seq analysis

Demultiplexing and alignment of sequencing read to the mouse transcriptome were performed using Cell Ranger software (version 3.0.2, 10X Genomics). We used the option "–forcecells 9000" in "cellranger count" to extract a larger number of cell barcodes from the trigeminal ganglion sample, as the automatic Cell Ranger estimate was inaccurate. Nuclei were used for clustering by Seurat[60]. The top 2000 genes were identified by variable feature selection based on a variance stabilizing transformation ("vst"). Then 50 principal components were utilized to calculate the *k*-nearest neighbors (KNN) graph based on the Euclidean distance in PCA space and the first 30 PCs were accordingly selected for the subsequent analysis according to the Jackstraw function. Resolution in the FindClusters function was set to 1.5. Clusters were then visualized using a Uniform Manifold Approximation and Projection (UMAP) plot. To annotate the cell types by gene markers, MAST differential gene expression analysis was performed by comparing nuclei in each cluster to the rest of the nuclear profiles. Genes with FDR < 0.05 and log fold change of 1 or more were selected as cell type markers.

## Integrative and interaction analysis of trigeminal ganglion and incisor

snRNA-Seq data from the trigeminal ganglion and scRNA-Seq data from the incisor were combined with Seurat and integration analysis was performed. RunPCA and RunUMAP were performed for further analysis.

CellChat[61] was used to explore the ligand–receptor interactions between trigeminal ganglion and incisor. The Seurat object was imported into CellChat followed by preprocessing functions to identifyOverexpressedGenes, identifyOverExpressedInteractions, and projectData with standard parameters set to analyze the potential cell–cell communication network. The core functions computeCommunProb, computeCommunProbPathway, and aggregateNet were run with standard parameters to infer the communication network and signaling pathway. NetAnalysis_contribution was run to compute the contribution of each ligand–receptor pair to the overall signaling pathway. NetVisual_circle, netAnalysis_signalingRole_heatmap, and netAnalysis_signalingRole_network were performed to analyze the senders and receivers.

## Retrograde tracing

The surgery was performed under anesthesia with 2% isoflurane and mice were aligned in a stereotactic frame (KOPF instruments, Tujunga, CA). Briefly, hair over the cheek was clipped, and the skin was aseptically prepared using alternating betadine and alcohol scrubs. Sagittal incisions were made on the skin and masseter muscle to expose the incisor. Two full-thickness defects of 0.3–0.4 mm diameter and 2.5–3.0 mm apart were created on the bone overlying the incisor mesenchyme using a microsurgical drill and a trephine drill bit. A 10 µl syringe (Hamilton Company, Reno, NV) coupled to a glass capillary pulled pipette was inserted into one of the bone defects and calking

material was applied around the glass pipette to form an airtight seal. 500 nl 2% CTB-Alexa488 (Molecular Probes: C-34775) was injected into the incisor mesenchyme over a period of 2 min. The glass pipette was kept in the bone defect for 15 min to allow the retrograde tracer to be fully absorbed by the dental pulp. The masseter muscle and cheek skin were then sutured closed. The mice were then subcutaneously injected with Buprenorphine SR (1 mg/kg) and allowed to recover on a heating pad until fully awake. Trigeminal ganglia were harvested up to 72 h after injection.

## Stereotaxic surgeries

Mice were anesthetized with isoflurane (induction 2.5%, maintenance 1.5%) and aligned in a stereotactic frame (KOPF instruments, Tujunga, CA). The skull was exposed under antiseptic conditions, and a small craniotomy was made with a thin drill. A total volume of 100 nl *Fgf1* shRNA lentivirus (Santa Cruz #SC-39445-V) or control shRNA (Santa Cruz, #SC-108080) was injected into the V3 region of the trigeminal ganglion (AP −2.0 mm. ML ± 2.4 mm, DV −6.3 mm) using a 10 µl syringe (Hamilton Company, Reno, NV) at a rate of 20 nl/min. Mice were euthanized 4 weeks after injection for examination.

## Denervation surgery

Microsurgery was performed to cut off the inferior alveolar nerve as described in a previous study[2]. The other side of the same mouse was operated following the same steps without damaging the nerve, which was the control.

## Incisor explant culture

The proximal end of the incisor was dissected from one-month-old *Gli1-LacZ* mice and cultured with a Trowell culture system in vitro. Briefly, the proximal end of the incisor was cultured in BGJb media supplementary with 10% fetal bovine serum (FBS), 1% penicillin/streptomycin (Invitrogen), and 0.1 mg/ml ascorbic acid (Sigma). For IAN implantation, the IAN was dissected out and surrounded by the proximal end. For neutralizing FGF1, we used Affi-Gel blue agarose beads (BioRad) and soaked them in FGF1 antibody (0.5 µg/ul, R&D system). Normal IgG control (0.5 µg/ul, R&D system) was used as control. For bead implantation, we used Affi-Gel blue agarose beads (BioRad) and soaked them in recombinant FGF1 protein (0.5 µg/µl, R&D system). BSA (0.5 µg/ul) was used as a control. The beads were put around the proximal end. Tissues were harvested after 3 days of culture and fixed in 4% paraformaldehyde.

## RNA sequencing

One week after tamoxifen induction, incisors from control and *Gli1^CreER;Fgfr1^fl/fl* mice were dissected. The proximal end of the incisor was collected, and RNA was extracted using an RNeasy Micro Kit (Qiagen, 74004). The quality of RNA samples was determined using an Agilent 2100 Bioanalyzer and all groups had RNA integrity (RIN) numbers > 9.0. For RNA-sequencing analysis, cDNA library preparation and sequencing were performed at the Technology Center for Genomics & Bioinformatics at the University of California, Los Angeles (UCLA), USA. The single read with 1 × 75 bp read length was performed on NextSeq500 High Output equipment for three pairs of samples from each group. Raw reads were trimmed, and aligned using Partek Flow with the mm10 genome, then normalized using the Upper quartile. Differential analysis was estimated by selecting transcripts with a significance of *P* < 0.05.

## Western blot analysis

Incisor proximal mesenchyme from control and *Gli1^CreER;Fgfr1^fl/fl* mice were collected, and tissues were incubated in RIPA buffer (Cell Signaling, 9806) with protease inhibitor (Thermo Fisher Scientific, 1861278) for 30 min at 4 °C. The soluble fraction was isolated with

centrifugation at 14,000×$g$ at 4 °C for 10 min. Total protein extracts were loaded in 4–15% precast polyacrylamide gel (Bio-Rad, 456-1084) and transferred to PVDF membranes (Millipore, ISEQ00005). Membranes were blocked with 5% non-fat dry milk for 1 h, then incubated with primary antibodies: anti-FGFR1 (Cell signaling technology 9740, 1:1000), anti-FGF1 (Abcam ab207321, 1:1000), anti-LC3 (1:1000, Abcam, ab48394), anti-p-ULK1 (Cell Signaling technology 14202, 1:1000), anti-p-mTOR (Cell Signaling technology 5536, 1:1000), anti-mTOR (Cell Signaling technology 2972, 1:1000), anti-p-p70S6K (Cell Signaling technology 9204, 1:1000), anti-p70S6K (Cell Signaling technology 2708, 1:1000), anti-p-S6 (Cell Signaling technology 4858, 1:1000), anti-S6 (Cell Signaling technology 2217, 1:1000), anti-p-Akt (Cell Signaling technology 4060, 1:1000), anti-Akt (Cell Signaling technology 9272, 1:1000), anti-p-p38 (Cell Signaling technology 4511, 1:1000), anti-p38 (Cell Signaling technology 8690, 1:1000), anti-p-ERK (Cell Signaling technology 4370, 1:1000), anti-ERK (Cell Signaling technology 4695, 1:1000), anti-p-JNK (Cell Signaling technology 9255, 1:1000), anti-JNK (Cell Signaling technology 9252, 1:1000), and anti-β-actin (Abcam ab20272, 1:1000) at 4 °C overnight. Proteins binding with primary antibodies were detected with horseradish-peroxidase (HRP)-conjugated secondary antibodies. The western blot images were detected by Azure 300 (Azure Biosystems).

## Co-immunoprecipitation

Proximal incisor mesenchyme from wild-type mice was collected and lysed in RIPA buffer. After pre-clearing using protein G Sepharose (GE Healthcare, 10280243), anti-FGF1 antibody (Abcam, ab207321), or Rabbit pre-immune IgG (Cell Signaling Technology, 3900) was added to the protein extract and rotated overnight at 4 °C. Then, protein G Sepharose beads were added and rotated at 4 °C for 2 h. Immune complexes were washed three times. Whole protein and immunoprecipitated protein were loaded and separated with 4–15% SDS–PAGE gels and transferred on PVDF membranes. After blocking with 5% non-fat dry milk, membranes were probed with anti-FGF1 (Abcam ab207321) or anti-FGFR1 (Cell Signaling Technology, 9740) antibodies, and signals were detected using Azure 300 (Azure Biosystems).

## Fluorescent double labeling

The use of fluorochromes in mineralization formation is widespread. We used calcein or Alizarin red to label the newly forming dentin in green or red fluorescence, respectively, to observe the dynamic dentin formation. Calcein was prepared in PBS at 4 mg/ml, and Alizarin red was dissolved in bacteriostatic water at 8 mg/ml, then filtered through a Millipore filter. We injected calcein i.p. 3 weeks after induction and injected Alizarin red 2 days before collecting the mice at a dosage of 20 mg/kg body weight. The fluorescent complex with calcium was detected with a fluorescence microscope.

## TUNEL assays

Cryosections were stained with a TUNEL assay kit (Click-iT™ Plus TUNEL Assay for In Situ Apoptosis Detection, Thermo Fisher Scientific, C10617) according to the manufacturer's protocol to detect cell apoptosis.

## Cell culture

MSCs were cultured as previously described[2]. Briefly, the proximal mesenchyme of the incisor was dissected out from control and $Gli1^{CreER};Fgfr1^{fl/fl}$ mice 7 days after induction and cut into pieces, then digested with collagenase type I. Cells were seeded into a culture plate after filtering with a strainer and cultured with α-MEM supplemented with 10% fetal bovine serum (FBS), 1% penicillin/streptomycin in 5% $CO_2$ at 37 °C.

For odontoblastic differentiation, MSCs were cultured in mineralization media (MM), which contained 10 mM β-glycerophosphate, 50 μg/ml ascorbic acid, and 10 nM dexamethasone.

## ROS assays

MSCs from control and $Gli1^{CreER};Fgfr1^{fl/fl}$ mice were seeded in a Millicell slide (Millpore), then treated with or without 100 ng/ml FGF1 for 24 h. For rapamycin treatment, cells were treated with or without 100 nM rapamycin for 24 h. 5 μM CellROX™ Green reagent (Invitrogen, C10444) was added into the complete media, then incubated for 30 min. Cells were washed with PBS and fixed with formaldehyde, then images were captured.

## Cell migration assay

We cultured MSCs from control and $Gli1^{CreER};Fgfr1^{fl/fl}$ mice and seeded cells in six-well plates, then wound healing assays were performed. A wound was created with a sterile plastic pipette tip on the monolayer cells. The cells were then incubated with 100 ng/ml recombinant FGF1 or 100 nM rapamycin in α-MEM for 12 and 24 h to allow migration back to the wound area. Cells were collected at 0, 12, and 24 h, fixed with 4% paraformaldehyde, and then stained with crystal violet (Sigma) to visualize migrated cells.

## Statistics and reproducibility

All statistical analysis was performed with GraphPad Prism and statistical data are presented as individual points and mean ± SD. Unpaired Student's $t$-test or one-way ANOVA analysis was used for comparisons, with $P < 0.05$ considered statistically significant. $N \geq 3$ for all samples. Each experiment was repeated independently three times.

## Reporting summary

Further information on research design is available in the Nature Portfolio Reporting Summary linked to this article.

## Data availability

Single nuclei RNA-sequencing (snRNA-Seq) and bulk RNA-seq datasets are available through the GEO database under accession code GSE197787. Source data are provided with this paper.

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

## Acknowledgements

We acknowledge Dr. Bridget Samuels for critical editing of the manuscript, USC Libraries Bioinformatics Service for assisting with data analysis, and the USC Office of Research and the USC Libraries for supporting our access to bioinformatics software and computing resources. This study was supported by funding from the National Institute of Dental and Craniofacial Research, National Institutes of Health (R01 DE025221 and R01 DE012711 to Y.C.).

## Author contributions

F.P. and Y.C. designed the study. F.P., L.M., J.J., J.F., Y.Y., T.G., X.H., J.L., J.H., and M.Z. carried out most of the experiments and data analyses. T.-V.H. participated in the microCT analysis. F.P., L.M., J.J., J.C., and Y.C. co-wrote the paper. Y.C. supervised the research.

## Competing interests

The authors declare no competing interests.
