## [Peer Review File · Nature Communications]

Sensory nerve niche regulates mesenchymal stem cell homeostasis via FGF/mTOR/autophagy axisREVIEWER COMMENTS

Reviewer #1 (Remarks to the Author):

In the manuscript by Pei et al, the authors investigate the role of nerves within the mouse incisor. High quality histological images show the pulp is primarily innervated by sensory nerves. Single cell sequencing was next used to identify FGF1 as a potential nerve signal that could regulate MSCs within the tooth environment. Researchers make use of in vivo and in vitro models, as well as biochemical and histological approaches to suggest this nerve-to-MSc axis regulates mTor mediated autophagy regulation. The addition of pharmacological treatments also strengthens the story. The major drawback lies in the indirect evidence supporting nerve-derived FGF1. There is an abundance of work on the role of FGF signaling in the teeth, the link between FGF and mTor signaling, the involvement of nerves in stem cell homeostasis and FGF ligands being produced by nerves. More recently, transgenic mouse models and pharmacological treatments to inhibit sensory nerves have been shown to regulate mesenchymal FGF signaling during bone development and following injury (PMID: 34663698, 34400627). This lack of definite connection detracts from the impact and novelty of this story.

Major Concerns

- While the similar phenotypes between sensory nerve ablation and Gli-CreER driven FGFR1 deletion are suggestive, a Advillin-CreER mediated deletion of FGF1 is required to support the author's claims. While the single cell data is suggestive, this data is biased by the overabundance of neural cells within the single cell (whereas within the in vivo dental pulp nerves are likely to be far less abundant). This assumption of crosstalk between nerves and MSCs via FGF signaling then leads to the indications of FGFR1 requirements. However, the abundance of other cell types producing other FGFs could have an equal or greater effect on MSC FGF signaling. As such, this nerve derived FGF signaling, a critical aspect to the novelty of this story, remains unsubstantiated.

-The CellChat experiments were poorly described in the text. Additional details regarding the identification of the FGF axis would be well placed. In addition, other signaling pathways identified through this technique should be investigated in the context of other ligands or pathways which have been proposed to be derived or regulated by sensory nerves in skeletal tissue. These additional pathways should be evaluated at the level of ligand and receptor expression in the single cell data. Undertaking a similar CellChat analysis and ligand-receptor expression comparison between different cell types within the dental pulp would also be beneficial to explore potential FGF signaling outside of neural inputs. The authors choice to force cell ranger to include more cells is poorly justified. Why was this altered? What additional cells were added as a result. If more cells were forced to be included, additional QC plots such as feature number, UMI number, % mito, % ribosomal, etc should be included to ensure these non-traditional changes did not have a significant effect on the results.

Minor issues

-FGF1 has previously been shown to be produced by the cervical loop epithelium of the incisor. A lack of positive control for the FISH technique done in Fig 2G,H makes assessing these images difficult (decalcifying the teeth can effect FISH which was likely not needed for the ganglion in E,F).

-The rigor of the story would be greatly improved by additional quantification (nerves/mm² in fig 1A, Dspp staining in Fig 1B, 3F, micro-CT of pulp space, western blots, etc)

-the spatial association between Gli1+ cells and nerves in Fig 1E is not overly convincing

-Sensory nerves have previously been implicated in regulating cell proliferation. Indeed, the authors own work supports these findings. Similarly, rapamycin treatment has also been

shown to regulate proliferation. Yet the authors attribute their phenotype mainly to autophagy. Additional work to uncouple proliferation and autophagy would be beneficial to enhance this aspect of the story.

-The discussion is fairly long and would benefit from the streamlining and reduction

-Figure S5 seemed critical for evaluating the authors claims and should receive consideration to be moved to main figures.

-

Reviewer #2 (Remarks to the Author):

The study focuses on the role of Fgf1 in dental stem cell niche homeostasis and differentiation of major dental cells types using continuously-growing mouse incisor as the model system. The most interesting aspect of a study includes the role of sensory neurons projecting from the trigeminal ganglion into dental stem cell niche. These sensory neurons, according to the results presented by the author team, produce, transport and release Fgf into the apical part hosting a stem cell niche of the continuously self-renewing mouse incisor. The paper is extraordinary solid, and promotes my enthusiasm for publishing it. The manuscript covers a wide array of experiments characterizing and proving the role of nerve-dependent release of Fgf1 in the mouse incisor tooth. Many of the key experiments and data appear hidden in the supplementary, which still might be improved by moving few panels to the main figures. Below I provide my comments and suggestions for improving the study:

1. It is of interest to understand the mechanism of Fgf1 transport and release from the nerve. Is it vesicular? Can Fgf1 be released only from nerve endings or throughout the entire length of the nerve fiber? This becomes an important question, because SHH and other ligands might be released in the same way and also in different locations. Do some sensory fibers terminate in a dental stem cell niche, or they always pass it by and terminate in the odontoblast layer within dentinal tubules? I would also suggest doing electron microscopy of the nerve fibers and labelling Fgf1 with immunogold antibodies. This is only a suggestion, and might be way beyond the opportunities to revise this manuscript.

2. Figure 2 and other main figures do not show the expression of Fgf1 in single cell data, and instead the authors moved it to the supplementary. I think this is wrong because the reader poorly connects what is in the main single cell figure and the rest of the manuscript logic. Single cell plots of Fgf1 and other ligands expression need to be presented in the main figure. It is important to discuss and show (in supplementary) which subtypes of sensory neurons express Fgf1, what is their molecular identity, transcription factor code, neurotransmitter specificity. Are they heterogeneous or rather homogeneous population? The major weakness of this paper is in the discussion of single cell analysis, which appears far incomplete.

3. It is necessary to check and rule out the other possible local sources of Fgf ligands by performing rather extensive RNAscope, HCR or in situ hybridization of section of a tooth together with dental follicle and surrounding tissues more extensively, as a single panel in Figure 2G does not look entirely convincing (there are some reddish cells mingling). I suggest doing it more accurately and extensively. It would be beneficial to check the available single cell datasets as well. Although the role of the nerve and Fgf1 nerve-dependent release is undeniably important, it might be good to present and discuss the other, possibly weaker alternative local sources of Fgf, if any. I just wish that the authors approached this part in a more convincing way.

Reviewer #3 (Remarks to the Author):

In this study, Pei et al. reported that sensory nerves promoted mesenchyme tissue homeostasis. They showed that sensory denervation led to mesenchymal stem cell loss and tissue disruption. Mechanically, sensory nerves signaled to MSCs via the FGF1-FGFR1 axis, which regulated retention, survival, and differentiation of MSCs in the adult mouse incisor. The findings could be of interest to those studying stem cell renewal and nerve-bone interaction. Specific questions include:

1. The sensory neurons positive for P92 significantly decreased in the trigeminal ganglion 7 days after tamoxifen induction in Advillin-CreER; RosaDTA mice as shown in Fig S1I-L, but they detected nerve fibers in the incisor one month after TMX induction in Fig S1M-P. Characterization of sensory nerve survival and the detection of FGF1 protein level within the same time frame will demonstrate the depletion efficiency as well as the production of FGF1 from sensory nerves. A statistical analysis during the time course would help demonstrate the dynamics, too.
2. The animal models used in the study were largely systematic including DTA induction in sensory neurons. An appropriate in vitro model may be used to illustrate the local effect of nerve action on mesenchymal tissue. Advillin-CreER target all the somatosensory neurons, and the neurotomy of trigeminal sensory branches could be an option to illustrate the function of a local sensory nerve to incisor.
3. The presence of an immunostaining signal against TH in the nerve subtypes is not sufficient to conclude that they are sympathetic since the sensory neuron subtype also expresses TH. A confirmation of the TH immunostaining signal in the Advillin-CreER; DTA mouse would be informative.
4. Instead of showing the nerve depletion in Fig S1, a side-by-side image of the immunostaining against sensory nerve fibers would help demonstrate the nerve depletion efficiency in Fig 1F.
5. The negative control should be included to show the signal specificity, such as in images in Fig 2E-L, Fig 3A, C, E, Fig S1A-H, and Fig S2-E-K.

October 26, 2022

NCOMMS-22-13295

Reviewer #1 (Remarks to the Author):

In the manuscript by Pei et al, the authors investigate the role of nerves within the mouse incisor. High quality histological images show the pulp is primarily innervated by sensory nerves. Single cell sequencing was next used to identify FGF1 as a potential nerve signal that could regulate MSCs within the tooth environment. Researchers make use of in vivo and in vitro models, as well as biochemical and histological approaches to suggest this nerve-to-MSC axis regulates mTor mediated autophagy regulation. The addition of pharmacological treatments also strengthens the story. The major drawback lies in the indirect evidence supporting nerve-derived FGF1. There is an abundance of work on the role of FGF signaling in the teeth, the link between FGF and mTor signaling, the involvement of nerves in stem cell homeostasis and FGF ligands being produced by nerves. More recently, transgenic mouse models and pharmacological treatments to inhibit sensory nerves have been shown to regulate mesenchymal FGF signaling during bone development and following injury (PMID: 34663698, 34400627). This lack of definite connection detracts from the impact and novelty of this story.

Major Concerns

- While the similar phenotypes between sensory nerve ablation and Gli-CreER driven FGFR1 deletion are suggestive, a Advillin-CreER mediated deletion of FGF1 is required to support the author's claims. While the single cell data is suggestive, this data is biased

by the overabundance of neural cells within the single cell (whereas within the *in vivo* dental pulp nerves are likely to be far less abundant). This assumption of crosstalk between nerves and MSCs via FGF signaling then leads to the indications of FGFR1 requirements. However, the abundance of other cell types producing other FGFs could have an equal or greater effect on MSC FGF signaling. As such, this nerve derived FGF signaling, a critical aspect to the novelty of this story, remains unsubstantiated.

RESPONSE: We thank the reviewer for this suggestion. We have performed two more experiments to confirm that sensory nerve-derived FGF1 is indispensable for MSC maintenance and mesenchymal homeostasis.

1. We neutralized FGF1 from sensory nerve with FGF1 antibody in incisor explant culture. The number of MSCs decreased with FGF1 neutralizing antibody-loaded beads surrounding the inferior alveolar nerve (IAN) (Fig. RL1). This supports our conclusion that the nerve-derived FGF1 is important for MSC maintenance.

Fig. RL1, A, Schematic of the incisor explant culture. B-C, Gli1⁺ cells in incisor explant with IAN surrounding the proximal end cultured with IgG loaded beads for 3 days. D-E,

Expression of Gli1⁺ cells in incisor explant with IAN surrounding the proximal end cultured with FGF1 antibody-loaded beads for 3 days. F, Quantification of Gli1⁺ cells in B and D.

2. Since there is no available *Fgf1^{fl/fl}* mouse model, we injected FGF1 shRNA lentivirus into V3 of the trigeminal ganglion, where the cell bodies of neurons innervating the incisor are localized (Fig. RL2A). *Fgf1* transcription was specifically and significantly inhibited in V3 of the trigeminal ganglion, while V1 and V2 showed no significant change (Fig. RL2B). Similar phenotypes were seen in FGF1 shRNA-treated mice and *Advillin-CreER;Rosa-DTA^{fl/fl}* mice. The dental pulp cavity was narrowed with abnormal dentin deposited in the proximal end of incisor in FGF1 shRNA-treated mice one month later (Fig. RL2C-a-j, D, E, F). The number of MSCs decreased in FGF1 shRNA-treated mice (Fig. RL2C-k-n, G). These demonstrated that the sensory nerve-derived FGF1 is crucial for the homeostasis of the mesenchyme and MSCs in the mouse incisor.

Fig. RL2, Decreased MSCs and disturbed mesenchymal homeostasis were detected after *Fgf1*-shRNA injection. A, The schematic of stereotaxic injection of *Fgf1*-shRNA. B, *Fgf1* decreased in V3 of trigeminal ganglion after *Fgf1*-shRNA injection. B-a-b, Expression of *Fgf1* in trigeminal ganglion with mock shRNA or *Fgf1*-shRNA. B-c-f, Expression of *Fgf1* and neurofilament in V3 of trigeminal ganglion with mock shRNA or *Fgf1*-shRNA. V1, V2 and V3 indicate the three branches of the trigeminal nerve. C-a-b, CT scanning of control and *Fgf1*-shRNA mice one month after injection. White arrowhead points to dental pulp cavity. C-c-f, Histological analysis of control and *Fgf1*-shRNA mice one month after injection. Yellow asterisk points to abnormal dentin formation. C-g-j, *Dspp* expression in control and *Fgf1*-shRNA mice. Yellow arrow points to the distance between the bending point of the cervical loop and the initiation of odontoblast differentiation. C-k-n, *Gli1*⁺ cells

in control and *Fgf1*-shRNA mice. D, Relative fluorescent intensity of *Fgf1*. E, Quantification of dental pulp cavity percentage in control and *Fgf1*-shRNA mice. F, Quantification of distance of *Dspp*⁺ cell to cervical loop in control and *Fgf1*-shRNA mice. G, Quantification of Gli1⁺ cells in control and *Fgf1*-shRNA mice. White dotted line outlines cervical loop. Scale bars, E-a-b, 1mm; D-a-b, 500 μ m; others, 100 μ m.

These results have been added in the Results (lines 241-264) in the revised manuscript.

We can capture partial nerve fibers in sagittal sections of the mouse incisor. To better visualize the nerve distribution easier to observe, we performed whole tissue clearing of the incisor and made a 3D view of the nerves in the adult mouse incisor. Abundant nerve fibers are enriched in the proximal end of incisor and go through the center of the incisor (Supplementary Movie 1 in revised manuscript).

-The CellChat experiments were poorly described in the text. Additional details regarding the identification of the FGF axis would be well placed. In addition, other signaling pathways identified through this technique should be investigated in the context of other ligands or pathways which have been proposed to be derived or regulated by sensory nerves in skeletal tissue. These additional pathways should be evaluated at the level of ligand and receptor expression in the single cell data. Undertaking a similar CellChat analysis and ligand-receptor expression comparison between different cell types within the dental pulp would also be beneficial to explore potential FGF signaling outside of neural inputs. The authors choice to force cell ranger to include more cells is poorly justified. Why was this altered? What additional cells were added as a result. If more cells were forced to be included, additional QC plots such as feature number, UMI number, %

mito, % ribosomal, etc should be included to ensure these non-traditional changes did not have a significant effect on the results.

RESPONSE: We appreciate the reviewer for these suggestions. In the top 20 signaling pathways involved in the interaction between trigeminal ganglion and incisor cells, we focused on the signaling secreted from sensory neurons in the trigeminal ganglion. It showed that MPZ, CADM, CNTN, FGF, L1CAM and THY1 were outgoing signals from trigeminal ganglion, of which MPZ, CADM, CNTN targeted the sensory neurons and glial cells. These signals therefore play an autocrine role in regulating nerves. FGF signaling from trigeminal ganglion showed that incoming signaling was received mainly by cell clusters in the incisor, such as proximal mesenchymal cells, dental follicle cells, TACs, epithelial cells and odontoblasts (shown in Supplementary Fig. 2A in revised manuscript). This suggested that FGF signaling was the most significant pathway in the interaction between the trigeminal ganglion and the incisor. Network analysis of FGF signaling further showed that sensory neurons in the trigeminal ganglion were the most prominent source of FGF ligands acting on proximal mesenchymal cells (MSC region) in the adult mouse incisor (shown in Fig. 2D in revised manuscript). We have added these details from our CellChat analysis about how the FGF axis was identified to the Results in the revised manuscript (lines 192-206). More details about CellChat were added to the Methods in the revised manuscript (lines 702-709).

In addition, we also detected other signaling pathways through which sensory nerves can regulate skeletal tissue, such as PDGF, VEGF, TGF β and WNT signaling^{1,2}. PDGF signaling was involved in the interaction between sensory nerves and incisor. The ligand PDGFA (*Pdgfa*) derived from sensory neurons can regulate proximal mesenchymal cells,

dental follicle cells, TACs and odontoblasts by acting on PDGFR α (*Pdgfra*) and PDGFR β (*Pdgfrb*) in mesenchymal cells in the incisor. The ligand from epithelial and endothelial cells in the incisor can also regulate these clusters (Fig. RL3A, C-D). VEGF signaling can also be secreted from the trigeminal ganglion. However, the main source of VEGF was glial cells, some from sensory neurons. The ligand VEGFA (*Vegfa*) didn't regulate mesenchymal cells in the incisor, whereas it targeted endothelial cells in the incisor by VEGFR1 (*Flt1*) and VEGFR2 (*Kdr*) (Fig. RL3B, C-D). TGF β and WNT signaling were not found involved in the interaction between TG and incisor in our study. The TGF β and WNT signaling involved in the cell-cell interaction in dental pulp of the incisor is shown in the CellChat analysis in clusters of dental pulp cells (Fig. RL4). This has been added to the Discussion (lines 499-512).

Fig. RL3 PDGF signaling participates in the interaction between sensory nerve and incisor. A, Overview of the inferred PDGF signaling pathway network. B, Overview of the inferred VEGF signaling pathway network. C, PDGF and VEGF ligand expression in trigeminal ganglion. D, Expression of PDGF receptors and VEGF receptors in adult mouse incisor.

We subsetted the dental pulp cell cluster and performed CellChat analysis according to the reviewer's suggestion. These clusters included the proximal mesenchymal cells (MSC region), TACs, dental follicle, dental pulp cells, pre-odontoblasts and odontoblasts. It showed that many signaling pathways are involved in the interaction, including WNT, FGF, BMP and TGF β . FGF signaling is from pre-odontoblasts, TACs and odontoblasts, which act on clusters in dental pulp. From FGF network analysis, TACs were the major source of FGF signaling, which predominantly targeted dental follicle and pulp cells (Fig. RL4). We have added this part to the Discussion in the revised manuscript (lines 533-540). We further detected the expression of FGF ligands in the adult mouse incisor (Fig. RL9).

Fig. RL4, A, Signaling involves in the interaction between different clusters of dental pulp in the adult mouse incisor. B, FGF signaling pathway network shows the main source of

FGF signaling in dental pulp is from pre-odontoblasts, TACs and odontoblasts, which predominantly act on dental follicle and pulp cells. C, Overview of the inferred FGF signaling pathway network.

We tried both the default parameter and the “force-cell 9000” parameter for data analysis. With the default parameters, Cellranger estimated 16,030 cells in the sample, which was far more than the actual cell number we loaded for library construction. Cellranger infers the cell number based on a barcode rank curve. Sometimes the ambient RNA leaked from the dead cells can lead to misjudgment of the cell number. Thus, we used the parameter force-cell 9000 following the suggestion from a technician at 10x Genomics. The QC plots show that the parameter we used did not have a significant effect on the results (Fig. RL5).

Fig. RL5. No significant difference between default and “force-cell 9000” parameters for data analysis.

Minor issues

-FGF1 has previously been shown to be produced by the cervical loop epithelium of the incisor. A lack of positive control for the FISH technique done in Fig 2G,H makes assessing these images difficult (decalcifying the teeth can effect FISH which was likely not needed for the ganglion in E,F).

RESPONSE: We thank the reviewer for the suggestion. *Fgf1* was expressed in the dental epithelium and follicle in the incisor during development (E16.5)³. In our study, we used an adult mouse model to study the tissue homeostasis, a model which exhibits some differences from the embryonic stage. For example, *Fgf7*, *Fgf16*, *Fgf17* and *Fgf18* can be

detected in embryonic stages, but cannot be detected in the adult mouse incisor (Fig.

Fig. RL6, Expression of *Fgf1* and *Fgf3* in adult mouse incisor. A, Feature plot *Fgf1* and *Fgf3* in adult mouse incisor. B, Expression of *Fgf1* in the incisor. C, *Fgf1* can't be detected in mesenchyme of incisor. D, *Fgf1* showed low expression in epithelium of incisor. E-F, Expression of *Fgf3* in the incisor. Scale bar, 100 μ m.

-The rigor of the story would be greatly improved by additional quantification (nerves/mm² in fig 1A, Dspp staining in Fig 1B, 3F, micro-CT of pulp space, western blots, etc)

RESPONSE: We have quantified our results and added them to the revised manuscript.

-the spatial association between Gli1+ cells and nerves in Fig 1E is not overly convincing

RESPONSE: We thank the reviewer for this suggestion. We have replaced the Fig. with a revised Fig. 1E to better show the spatial association between Gli1⁺ cells and nerves.

-Sensory nerves have previously been implicated in regulating cell proliferation. Indeed, the authors own work supports these findings. Similarly, rapamycin treatment has also been shown to regulate proliferation. Yet the authors attribute their phenotype mainly to autophagy. Additional work to uncouple proliferation and autophagy would be beneficial to enhance this aspect of the story.

RESPONSE: We agree with the reviewer. In our study, we found that autophagy mainly played a role in resisting apoptosis to sustain the MSCs. The dynamic turnover of the incisor is driven by MSCs giving rise to TACs, which then differentiate into odontoblasts to generate dentin. The number of MSCs was the primary change in the incisor in our study, which then led to decreased TACs. We used Ki67 as a marker to label the TACs. We demonstrated the decreased MSCs could be restored with autophagy activation and

apoptosis resistance, so we used Ki67 to check whether TACs were also rescued. We have changed “Ki67+ cells” to “Ki67+ TACs” in the revised manuscript. We also made the statement clearer in the result in revised manuscript (lines 437-439; 473-475).

-The discussion is fairly long and would benefit from the streamlining and reduction

RESPONSE: We have streamlined and revised the discussion.

-Fig. S5 seemed critical for evaluating the authors claims and should receive consideration to be moved to main Fig.s.

RESPONSE: We thank the reviewer for this suggestion. Supplementary Fig. 5 (Supplementary Fig. 6 in revised manuscript) was a supplement to Fig.5 (in revised manuscript). Fig. 5A-J and Supplementary Fig. 6A-M demonstrated that decrease in MSCs led to a subsequent reduction in the TAC population after loss of FGF signaling. Fig. 5K-O and Supplementary Fig. 6N-R demonstrated that odontoblast differentiation was impaired. Fig. 5P-R and Supplementary Fig. 6R-X demonstrated that abnormal dentin formation was caused by slower migration. Fig. 5 and Supplementary Fig. 6 showed the same topic that FGF signaling is important for MSC regulation in adult mouse incisor, which influences MSC maintenance, TAC number, and odontoblast differentiation and migration.

Finally, regarding the two references on how sensory nerves may regulate bone development and injury repair (PMID: 34663698, 34400627), we have added them in our introduction and discussion (Page5 and 24). However, our study reveals how sensory nerve secreted FGF1 directly act upon MSCs to regulate adult tissue homeostasis. This

important finding significantly advances our understanding of the molecular regulating mechanism of MSCs in adult tissue homeostasis.

Reviewer #2 (Remarks to the Author):

The study focuses on the role of Fgf1 in dental stem cell niche homeostasis and differentiation of major dental cells types using continuously-growing mouse incisor as the model system. The most interesting aspect of a study includes the role of sensory neurons projecting from the trigeminal ganglion into dental stem cell niche. These sensory neurons, according to the results presented by the author team, produce, transport and release Fgf into the apical part hosting a stem cell niche of the continuously self-renewing mouse incisor. The paper is extraordinary solid, and promotes my enthusiasm for publishing it. The manuscript covers a wide array of experiments characterizing and proving the role of nerve-dependent release of Fgf1 in the mouse incisor tooth. Many of the key experiments and data appear hidden in the supplementary, which still might be improved by moving few panels to the main Figures. Below I provide my comments and suggestions for improving the study:

- 1. It is of interest to understand the mechanism of Fgf1 transport and release from the nerve. Is it vesicular? Can Fgf1 be released only from nerve endings or throughout the entire length of the nerve fiber? This becomes an important question, because SHH and other ligands might be released in the same way and also in different locations. Do some sensory fibers terminate in a dental stem cell niche, or they always pass it by and terminate in the odontoblast layer within dentinal tubules? I would also suggest doing*

electron microscopy of the nerve fibers and labelling Fgf1 with immunogold antibodies. This is only a suggestion, and might be way beyond the opportunities to revise this manuscript.

RESPONSE: We appreciate the reviewer for these suggestions. To better visualize the spatial distribution of nerves and nerve terminals in the adult mouse incisor, we transparentized the incisor and showed the nerve fibers in 3D (shown in Movie S1 in revised manuscript). Abundant nerve fibers were detected in the proximal end of the incisor. In the proximal region, nerve fibers were located in the center of the incisor, but were not found in the odontoblast layer (Fig. RL7 and Supplementary Movie 1). Some nerve fibers terminated in the proximal region of the incisor, while some went through the center of the incisor (Fig. RL7 and Supplementary Movie 1). This was added to the Results in the revised manuscript (lines 125-129). CGRP, a peptide transmitter, is synthesized in neurons and packaged into vesicles for transport to axon terminals. Then the presynaptic terminals take the form of focal swelling (called axonal varicosities) to release CGRP from its storage vesicles⁴. FGF1 from the sensory nerve may have a similar release mechanism, which needs further study.

Fig. RL7, Wholemout staining of neurofilament in the proximal region of adult mouse incisor. Scale bar 50 μ m.

2. Figure 2 and other main Figures do not show the expression of Fgf1 in single cell data, and instead the authors moved it to the supplementary. I think this is wrong because the reader poorly connects what is in the main single cell Figure and the rest of the manuscript logic. Single cell plots of Fgf1 and other ligands expression need to be presented in the main Figure. It is important to discuss and show (in supplementary) which subtypes of sensory neurons express Fgf1, what is their molecular identity, transcription factor code, neurotransmitter specificity. Are they heterogeneous or rather homogeneous population? The major weakness of this paper is in the discussion of single cell analysis, which appears far incomplete.

RESPONSE: We appreciate the reviewer for these suggestions. We have moved the *Fgf1* and other ligand expression to the main Figure.

In the trigeminal ganglion, there are different cell types of sensory neurons, which include thermal nociceptors (*Trpv1* and *Trpm8*), mechanoreceptors (*Gfra2* and *Pou4f2*), peptidergic nociceptors (*Tac1*), non-peptidergic nociceptors (*Cd55* and *Scn11a*) and so on⁵. We checked the expression patterns and the colocalization of these markers with *Fgf1*. Interestingly, it showed that *Gfra2*⁺ sensory neurons for mechanosensation have similar expression of *Fgf1* and partially colocalized with *Fgf1* (Fig. RL8A). When we verified this *in vivo*, most sensory neurons innervating the incisor (CTB-488⁺ neurons) and secreting *Fgf1* (*Fgf1*⁺ neurons) expressed *Gfra2* (CTB-488⁺/*Fgf1*⁺/*Gfra2*⁺ sensory neurons) (Fig. RL8B-I). This suggested that FGF1 was secreted from the sensory neurons responsible for mechanosensation. We have added these in the Results (line231-236) in the revised manuscript.

Fig. RL8, Mechanosensation neurons innervating incisor secrete FGF1. A, Feature plot of *Fgf1* and *Gfra2* in trigeminal ganglion. B, F, Colocalization of CTB-488, *Gfra2* and *Fgf1* in V3 of trigeminal ganglion. C, G, CTB-488⁺ neurons in trigeminal ganglion. D, H, Expression of *Gfra2* in trigeminal ganglion. E, I, Expression of *Fgf1* in trigeminal ganglion. Yellow arrows indicate CTB-488⁺/*Gfra2*⁺/*Fgf1*⁺ neurons; yellow arrowheads indicate CTB-488⁺/*Fgf1*⁺ neurons. Scale bars, 100 μ m.

3. It is necessary to check and rule out the other possible local sources of Fgf ligands by performing rather extensive RNAscope, HCR or in situ hybridization of section of a tooth together with dental follicle and surrounding tissues more extensively, as a single panel in Fig. 2G does not look entirely convincing (there are some reddish cells mingling). I suggest doing it more accurately and extensively. It would be beneficial to check the available single cell datasets as well. Although the role of the nerve and Fgf1 nerve-dependent release is undeniably important, it might be good to present and discuss the other, possibly weaker alternative local sources of Fgf, if any. I just wish that the authors approached this part in a more convincing way.

RESPONSE: We thank the reviewer for these suggestions. We agree there is local FGF signaling, which can regulate dental mesenchyme and epithelium. Previous studies showed that the FGF signaling (FGF3 and FGF10) in the mesenchyme is crucial for the dental epithelium by acting on FGFR1 and FGFR2 in the epithelium^{3,6}. In our study, apart from the main source of FGF signaling from sensory nerves, some FGF signaling from TACs and odontoblasts in the incisor appeared to play an auxiliary role in mesenchymal cells and epithelium in the incisor (shown in Fig. 2E in revised manuscript). According to the reviewer's suggestion, we detected FGF ligands in the adult mouse incisor using

scRNAseq and *in vivo* to check the local FGF signaling (Fig. RL9). Sparse *Fgf1* was detected in epithelium. *Fgf3*, *Fgf8* and *Fgf10* were expressed in dental mesenchymal cells. Expression of *Fgf9* was found in the epithelium (Fig. RL9). We further analyzed the interaction between different clusters in dental pulp. Based on FGF network analysis, TACs were the major source of FGF signaling, which predominantly targeted dental follicle and pulp cells (Fig. RL3). This part was added to the Results (lines 211-219 and Supplementary Fig. 3) and Discussion (lines 533-540) in the revised manuscript.

Fig. RL9, Different *Fgf* ligands in adult mouse incisor. A-B. The expression of different *Fgf* ligands plotted from scRNAseq of adult mouse incisor. C-D. The expression of *Fgf3* in the incisor. E-F. The expression of *Fgf8* in the incisor. G-H. The expression of *Fgf9* in the

incisor. I-J. The expression of *Fgf10* in the incisor. White dotted line outlines cervical loop.
Scale bars, 100 μ m.

Reviewer #3 (Remarks to the Author):

In this study, Pei et al. reported that sensory nerves promoted mesenchyme tissue homeostasis. They showed that sensory denervation led to mesenchymal stem cell loss and tissue disruption. Mechanically, sensory nerves signaled to MSCs via the FGF1-FGFR1 axis, which regulated retention, survival, and differentiation of MSCs in the adult mouse incisor. The findings could be of interest to those studying stem cell renewal and nerve-bone interaction. Specific questions include:

1. The sensory neurons positive for P92 significantly decreased in the trigeminal ganglion 7 days after tamoxifen induction in Advillin-CreER; RosaDTA mice as shown in Figure S1I-L, but they detected nerve fibers in the incisor one month after TMX induction in Figure S1M-P. Characterization of sensory nerve survival and the detection of FGF1 protein level within the same time frame will demonstrate the depletion efficiency as well as the production of FGF1 from sensory nerves. A statistical analysis during the time course would help demonstrate the dynamics, too.

RESPONSE: We thank the reviewer for these suggestions. We detected the sensory nerve survival in the trigeminal ganglion, nerve fibers and FGF1 expression in the incisor within the same time frame (one month after tamoxifen induction). We replaced the previous Figures with Fig. 1E and Supplementary Fig. 1B, and revised the language in the Results (lines 145-149). It showed the decreased sensory neurons in the trigeminal ganglion and nerve fibers in the incisor with the decreased FGF1 in the incisor in *Advillin-CreER;Rosa-DTA* mice (Fig. RL10).

Fig. RL10, Decreased sensory neurons in trigeminal ganglion with impaired nerve fibers and FGF1 in the incisor of *Advillin-CreER;Rosa-DTA* mice one month after tamoxifen induction. A-a-d, Sensory neurons stained with P92 in the trigeminal ganglion of control

and *Advillin-CreER;Rosa-DTA* mice one month after tamoxifen induction. A-e-f, Nerves fibers labeled with neurofilament in control and mutant mice. A-g-j, Expression of FGF1 decreased in the incisor one month after sensory nerve deletion. B, Quantification of neuron numbers in trigeminal ganglion in control and *Advillin-CreER;Rosa-DTA* mice. C, Quantification of nerve fibers in control and mutant mice. D, Quantification of relative fluorescent intensity of FGF1 in control and mutant mice. White dotted line outlines cervical loop. Scale bars, 100 μ m.

2. The animal models used in the study were largely systematic including DTA induction in sensory neurons. An appropriate in vitro model may be used to illustrate the local effect of nerve action on mesenchymal tissue. Advillin-CreER target all the somatosensory neurons, and the neurotomy of trigeminal sensory branches could be an option to illustrate the function of a local sensory nerve to incisor.

RESPONSE: We appreciate the reviewer for these suggestions. We performed the denervation by cutting the inferior alveolar nerve (IAN), which is the branch of the trigeminal nerve responsible for the innervation of the lower incisor. We found that the number of MSCs decreased after denervation (Fig. RL11), which suggested the local effect of sensory nerve acting on mesenchymal tissue. We further used incisor explant culture as an *in vitro* model to study the local effect of sensory nerve on MSCs (in Supplementary Fig.4A in revised manuscript). To verify that sensory nerve-derived FGF1 is crucial for MSCs, we used anti-FGF1 antibody to neutralize FGF1 from IAN with incisor explant culture. It showed that the number of MSCs decreased with FGF1 neutralizing antibody-loaded beads surrounding the IAN. This further supports that the nerve-derived

FGF1 is important for MSC maintenance (Fig. RL1). These have been added in Fig. 3A and described in the Results (lines 242-252) in the revised manuscript.

Fig. RL11, Denervation of the inferior alveolar nerve leads to decreased MSCs in incisor. A, Expression of neurofilament and β -gal in sham and denervation group after one month denervation. B, Quantification of nerve fibers in sham and denervation group. C, Quantification of Gli1⁺ cells in sham and denervation group. Scale bars, 100 μ m.

3. *The presence of an immunostaining signal against TH in the nerve subtypes is not sufficient to conclude that they are sympathetic since the sensory neuron subtype also expresses TH. A confirmation of the TH immunostaining signal in the Advillin-CreER; DTA mouse would be informative.*

RESPONSE: We thank the reviewer for the suggestion. We performed TH staining in *Advillin-CreER; Rosa-DTA* mice according to the reviewer's suggestion. It showed similar

signal pattern to that of *Advillin-CreER;Rosa-DTA* mice, which showed a very limited number of TH⁺ fibers in the proximal region of the incisor one month after tamoxifen induction (Fig. RL12).

Fig. RL12, Expression of TH in *Rosa-DTA* mice and *Advillin-CreER;Rosa-DTA* mice one month after tamoxifen induction. Scale bar, 100 μ m.

4. Instead of showing the nerve depletion in Figure S1, a side-by-side image of the immunostaining against sensory nerve fibers would help demonstrate the nerve depletion efficiency in Figure 1F.

RESPONSE: We thank the reviewer for these suggestions. We have performed immunostaining with neurofilament to label nerve fibers and β -gal to label Gli1⁺MSCs in

Gli1-LacZ mice and *Advillin-CreER;Rosa-DTA;Gli1-LacZ* mice one month after tamoxifen induction. It showed that nerves were depleted efficiently, and the number of MSCs decreased. We have replaced the previous Figures with a revised Fig.1 E in the revised manuscript (lines 176-180).

5. The negative control should be included to show the signal specificity, such as in images in Fig 2E-L, Fig 3A, C, E, Fig S1A-H, and Fig S2-E-K.

RESPONSE: We thank the reviewer for these suggestions. We have included negative controls to show the signal specificity.

Fig. RL12, Negative control in trigeminal ganglion, incisor and cells. A-D, *Fgfr1* and negative control in *tdTomato*⁺ cells. E-F, *Fgf1* and negative control in trigeminal ganglion.

G-H, FGF1 and negative control in trigeminal ganglion. I-J, FGF1 and negative control in incisor. K-L, FGFR1 and negative control in incisor. White and black arrow indicate signals. White and black dot line indicate neurons. Scale bar, A-D, 50 μ m; others, A-D, 100 μ m.

We greatly appreciate all of the insightful suggestions offered by the reviewers and feel that our revised manuscript has greatly improved through this review process. Thank you very much.

Sincerely,

Yang Chai, DDS, PhD
University Professor CCMB, USC

References

- 1 Tower, R. J. *et al.* Spatial transcriptomics reveals a role for sensory nerves in preserving cranial suture patency through modulation of BMP/TGF-beta signaling. *Proc Natl Acad Sci U S A* 118, doi:10.1073/pnas.2103087118 (2021).
- 2 Lee, S. *et al.* NGF-TrkA signaling dictates neural ingrowth and aberrant osteochondral differentiation after soft tissue trauma. *Nat Commun* 12, 4939, doi:10.1038/s41467-021-25143-z (2021).
- 3 Porntaveetus, T. *et al.* Expression of fibroblast growth factors (Fgfs) in murine tooth development. *J Anat* 218, 534-543, doi:10.1111/j.1469-7580.2011.01352.x (2011).
- 4 Edvinsson, L., Haanes, K. A., Warfvinge, K. & Krause, D. N. CGRP as the target of new migraine therapies - successful translation from bench to clinic. *Nat Rev Neurol* 14, 338-350, doi:10.1038/s41582-018-0003-1 (2018).
- 5 Yang, L. *et al.* Human and mouse trigeminal ganglia cell atlas implicates multiple cell types in migraine. *Neuron* 110, 1806-1821 e1808, doi:10.1016/j.neuron.2022.03.003 (2022).
- 6 Harada, H. *et al.* Localization of putative stem cells in dental epithelium and their association with Notch and FGF signaling. *J Cell Biol* 147, 105-120, doi:10.1083/jcb.147.1.105 (1999).

REVIEWERS' COMMENTS

Reviewer #1 (Remarks to the Author):

Congratulations on a nice story.

Some things that should be fixed before publication:

-The authors still overstate in the intro and discussion that this is first description of sensory nerves on MSCs. There are actually many papers on this in other contexts such as the digit tip, limb regeneration, mandible fracture and heterotopic ossification. Their work is the first to define in the incisor so they should focus there.

-Authors should cite and discuss literature on role of nerve on digit tip regeneration as well as mandible fracture. PMID: 31509739 PMID: 24958860 PMID: 31285319 PMID: 27376984

-I think they overstate the results in Fig RL1 as this really doesn't do anything to distinguish nerve vs other cell sources of FGF1.

- I think Fig RL2 is much more conclusive and believable in terms of establishing a direct nerve FGF to MSC connection.

-Fig RL7 doesn't really address the reviewers comment about where FGF1 could be released (terminal vs throughout axon, free vs vesicular).

Reviewer #2 (Remarks to the Author):

The authors addressed all my comments pretty well. I am happy with their response, and suggest to proceed with publishing this study.

Reviewer #3 (Remarks to the Author):

The revised version has addressed the primary concern in cellular specificity of Fgf1 and has provided the control experiments needed.

December 12, 2022

NCOMMS-22-13295A

Response to Reviewer #1's comments:

Some things that should be fixed before publication:

-The authors still overstate in the intro and discussion that this is first description of sensory nerves on MSCs. There are actually many papers on this in other contexts such as the digit tip, limb regeneration, mandible fracture and heterotopic ossification. Their work is the first to define in the incisor so they should focus there.

We changed our previous statement in the discussion from “This highlights a previously unknown role of sensory nerves in regulating stem cells to maintain tissue homeostasis...” to “This highlights an important role of sensory nerves in regulating stem cells to maintain tissue homeostasis..” There are no other places where we stated that our study is the first description of sensory nerves on MSCs.

-Authors should cite and discuss literature on role of nerve on digit tip regeneration as well as mandible fracture. PMID: 31509739 PMID: 24958860 PMID: 31285319 PMID: 27376984

We have cited these publications in our revised manuscript and discussed some of them as they are relevant to our study.

-I think they overstate the results in Fig RL1 as this really doesn't do anything to distinguish nerve vs other cell sources of FGF1.

- I think Fig RL2 is much more conclusive and believable in terms of establishing a direct nerve FGF to MSC connection.

-Fig RL7 doesn't really address the reviewers comment about where FGF1 could be released (terminal vs throughout axon, free vs vesicular).

We appreciate the reviewer's comments and the opportunity to provide further clarifications. Regarding the source of FGF1 and its regulatory role on MSC, we provide an array of evidence based on both in vivo and in vitro studies. In our in vivo studies, we used Fgf1 shRNA lentiviral injection into V3 (third division) of the trigeminal nerve, which led to the decrease of MSCs and dentin defect similar to the ones in *Advillin-CreER;Rosa-DTA* mice. Our in vivo denervation study also resulted in a reduction of FGF1. In our in vitro incisor explant culture study, FGF1 neutralizing antibody treatment resulted in reduction of MSCs whereas exogenous FGF1 treatment can help to maintain MSCs in the incisor explant. Finally, loss of *Fgfr1* in MSCs resulted in a compromised FGF signaling and tissue homeostasis. Collectively, these studies support the conclusion that FGF1 from the trigeminal nerve regulates MSCs and incisor tissue homeostasis.

Reviewer #2:

We thank the reviewer for the insightful comments and suggestions

Reviewer #3:

We thank the reviewer for the insightful comments and suggestions

We greatly appreciate all the constructive suggestions offered by you and the reviewers and feel that our revised manuscript has greatly improved through this review process. Thank you very much for your help.

With best wishes,

Yang Chai, DDS, PhD
University Professor
George & MaryLou Boone Chair
in Craniofacial Molecular Biology
Center for Craniofacial Molecular Biology
University of Southern California
2250 Alcazar Street, CSA 103
Los Angeles, CA 90033
e-mail: ychai@usc.edu
phone: (323) 442-3480